# Emergent patterns of patchiness differ between physical and planktonic properties in the ocean

Patrick Clifton Gray [1,2] ✉, Emmanuel Boss [1], Guillaume Bourdin [1], Mission Microbiomes AtlantECO*, Tara Pacific Consortium*, Mission Microbiomes CEODOS Chile consortium* & Yoav Lehahn [2] ✉

While a rich history of patchiness research has explored spatial structure in the ocean, there is no consensus over the controls on biological patchiness and how physical-ecological-biogeochemical processes and patchiness relate. The prevailing thought is that physics structures biology, but this has not been tested at basin scale with consistent in situ measurements. Here we use the slope of the relationship between variance vs spatial scale to quantify patchiness and ~650,000 nearly continuous ($dx \sim 200$ m) measurements - representing the Atlantic, Pacific, and Southern Oceans - and find that patchiness of biological parameters and physical parameters are uncorrelated. We show variance slope is an emergent property with unique patterns in biogeochemical properties distinct from physical tracers, yet correlated with other biological tracers. These results provide context for decades of observations with different interpretations, suggest the use of spatial tests of biogeochemical model parameterizations, and open the way for studies into processes regulating the observed patterns.

Most phytoplankton species near the ocean's surface have a doubling time on the order of a day[1]. This rapid doubling time means phytoplankton quickly respond to changes in external forcings[2]. Coupled with the dynamic physical nature of the marine environment and grazing pressures with resulting loss rates similar to growth rates for phytoplankton[3,4], the ocean has considerable spatial heterogeneity, or patchiness, at all scales[5–8]. This spatial heterogeneity supports increased diversity by creating different environmental conditions within close proximity, ecosystem stability through diversity and maintenance of seed populations, and efficient transfer of carbon up trophic levels[9]. The link between patchiness and the efficient transfer of carbon in the marine ecosystem is largely due to dense aggregations that make it energy efficient for zooplankton to graze, and patches of zooplankton make it feasible for planktivorous fish to meet their own energy requirements[10–12]. This spatial variability of phytoplankton and zooplankton, and the generally increasing patchiness as one moves up the trophic chain, has

important implications for food availability and productivity of the entire ocean food web[13–15].

The spatial patterns of the plankton ecosystem and their drivers have long been investigated. An early outline by Hutchinson discusses nutrient and temperature gradients, stochastic events such as storms, intra-species signaling, competition, and predation among processes that likely control these plankton patterns[16]. An updated synthesis by Levin described pattern and scale as the central problem in ecology, arguing for cross-scale investigations to understand the mechanisms and consequences of marine patchiness[5]. The scale-dependent spatial heterogeneity exhibited by both biology and physics has been a central focus of studies into pattern and scale in the ocean[6], with much of this work attempting to divide spatial scales into domains of scale-independence and then investigating the dominant processes influencing biology in those domains[17,18]. Power spectral analysis, a Fourier decomposition where variance is partitioned into the contribution of bands of specific frequencies or wavenumbers (see[19] for an early review

[1]School of Marine Sciences, University of Maine, Orono, ME, USA. [2]Department of Marine Geosciences, Charney School of Marine Sciences, University of Haifa, Haifa, Israel. *Lists of authors and their affiliations appears at the end of the paper. ✉e-mail: patrick.gray@maine.edu; ylehahn@univ.haifa.ac.il

on the implementation of the technique in ecology), is often used to describe phytoplankton patchiness and quantify this scale-dependence. The scaling behavior of a geophysical variable is typically represented as a log-log plot of variance against length scale. If that relationship is approximately linear (in log space) this represents a power law relation between variance and spatial scale where the exponent of that power law relationship is referred to as 'the spectral slope'[20]. A flatter spectral slope indicates less of a decrease of variance with scale, i.e. more variability at smaller scales relative to larger scales, when compared to a steeper slope. Spectral slope has often been used synonymously with patchiness, but can also be interpreted as the cascade of variance across scales. While there are a range of issues with this analysis, such as discarding of phase information[9,21], it is commonly used as an indicator of the distribution of spatial variance.

There is over a century of inquiry into the spatial heterogeneity of ocean physics. Energy in the ocean generally cascades down spatial scales. Mesoscale eddies spin off of large-scale circulation features. Energy then transfers from these major eddies to smaller turbulent whorls and finally to a small enough scale that energy is dissipated by viscosity into heat[22–24]. This turbulent cascade generally has a spectral slope of -5/3 in the inertial subrange (3D and a spatial range smaller than the major energy containing eddies, but larger than viscous eddies). While Kolomogrov initially proposed this cascade of variance for fully developed 3D turbulence in the ocean's inertial subrange, O(10-0.001) meters depending on flow characteristics[25], the -5/3 spectral slope has been shown to hold generally true for the inertial subrange of 2D ocean turbulence O(100-1) km as well[26]. Temperature generally follows the same cascade as turbulent kinetic energy. It has a source of variance at large scales, a dissipation of variance at small scales, and it is typically being mixed by similar turbulent processes[27], though could have a variance source at smaller scales such as spatially heterogenous upwelling of cooler water.

The patchiness of phytoplankton was initially thought to be linked to turbulent stirring simply by physical processes[28], with a general conclusion of consistent scaling behavior between physical and biological features[18]. This conclusion was drawn largely from its spectral slope that appeared similar to the -5/3 of 2D turbulence[28,29]. Modeling studies indicated that the spectral slope of phytoplankton was also influenced by the phytoplankton reproductive rate, diagnosed by a scaling break at larger spatial scales where the spectral slope became flatter than that of a passive tracer[30,31]. This led to a substantial amount of work, with many conflicting results, investigating the time and length scales where the contribution of biologically generated spatial variability (growth rates, grazing, etc) to the total planktonic spatial variability is significant vs where physics dominates[6,7,9,25,32].

Abrahams[7] suggested, and others have come to a similar conclusion via modeling[33], that the spatial patterns in both plankton and physical tracers are a product of the timescales of their response to physically driven changes. For example, the equilibration of temperature to a heat flux is considerably slower than the response of phytoplankton to a nutrient flux. A small burst of cold nutrient-rich water thus leads to increased spatial heterogeneity in chlorophyll-a (chl-a) vs temperature, but as nutrients are exhausted and the temperature equilibrates over time the patchiness in chl-a and temperature would return to covarying[32]. Modeling work shows a steeper spectral slope in temperature compared to chl-a, attributing some of the slope variability to low frequency physical variations and the correlation of physical and biological variables to "the dominant role of physics in the spatial variability of plankton distribution"[34].

Despite these theoretical advances, interpretations from observational studies remain inconsistent. Some work concludes that at <1 km scales biological processes are important for patch dynamics[10,35], while other studies assert that at this scale spatial distributions of phytoplankton are dominated by turbulent diffusion. For example, this latter conclusion is advanced by an investigation that found a strong correlation between chl-a and temperature patchiness at scales below 5 km yet no correlation at scales from 5 km to 80km[19]. Instances spring up where phytoplankton are patchier than zooplankton[36], conflicting findings that the patchiness of phytoplankton is between that of zooplankton and physical factors[7,15,28]. Work assessing the spectral slope of chl-a in the same region for multiple years found variable slopes from month to month[25,37]. After decades of work, it is not clear at which spatiotemporal scales biological processes are important, or even which processes are dominant in controlling patchiness in the ocean.

While biophysical ocean patchiness has been relatively well documented on local scales, its characteristics across the globe have not yet been analyzed. Therefore, there is little data to test new theories, constrain impacts on productivity and diversity, and simply describe the patchiness itself. Importantly, this limits our ability to parameterize global biogeochemical models that are computationally limited from explicitly modeling these scales in our global models of the carbon cycle.

Here we address this gap, and explore the global characteristics of biophysical patchiness using a unique dataset of in situ surface plankton-related optical properties from the S/V *Tara* program. All data comes from the Tara Pacific[38] and Tara Microbiome missions which represent five years of observations spanning the Atlantic, Pacific, and Southern Oceans. This data was collected and processed consistently. After conservative quality control this dataset amounts to $N = 661,552$ ($dt = 1$ minute, $dx ∼ 200$ m) measurements used in this analysis (Fig. 1). As we aim to discern between patterns of physical and biological patchiness, and specifically to test if patchiness in biology is explained by patchiness in physics, the measurements are divided into two corresponding groups of variables, with the physical domain being represented by temperature, salinity and density, and the biological domain by chl-a, particulate attenuation of light at 443 nm ($c_p(443)$), and an optical proxy for mean particle size ($\gamma$)[39]. The latter two, because they represent other living (e.g. bacteria and micro-grazers) and non-living particles in addition to phytoplankton, are referred to as biogeochemical properties.

## Results and discussion

While the Fourier-based spectral slope is a useful and commonly used tool for characterization of patchiness, it is not suitable for analyzing unevenly sampled datasets. To overcome this limitation here we quantify patchiness via the dependence of the variance, $V$, on length scale, $L$ following[33]. As found in previous work, $V$ varies mostly linearly with $L$ in log space (Fig. 2), indicating a power law relationship of the form: (1) $V = L^\Gamma$, where $\Gamma$ describes the slope (exponent) of the relationship in log space. In this work we focus on $\Gamma$, the "variance slope". As with a power spectral density slope, a transect with a lower $\Gamma$ would have relatively more variance retained at smaller length scales relative to larger scales compared to a transect with higher $\Gamma$, i.e. a lower $\Gamma$ is patchier. This metric can be thought of qualitatively as a ratio of large scale to small-scale variance (Figure S1), with $\Gamma$ increasing as the large-scale variance increasingly dominates the small. This parameter is similar and strongly correlated with the conventional Fourier-based spectral slope exponent (Figures S2 and S3), but our approach (from[33]) has the advantage of being applicable to data with gaps, as long as they are smaller than 20% of a transect. This parameter is non-parametric and the value is the same for a transect before and after a log-transformation, allowing robust comparison between values like chlorophyll-a, which is often log-distributed[40], and values like temperature and salinity. Based on the observed relationship between $\Gamma$ and the power spectral density slope (Figure S3) the classic -5/3 slope found in previous studies for the inertial subrange would correspond to a $\Gamma$ value of 0.84.

Specifically in this work, variance was calculated on subsets of 500 samples over a set of 11 log-distributed windows from 3 to 500 samples, spanning ∼0.6 km to 100 km. Both variance and window size were

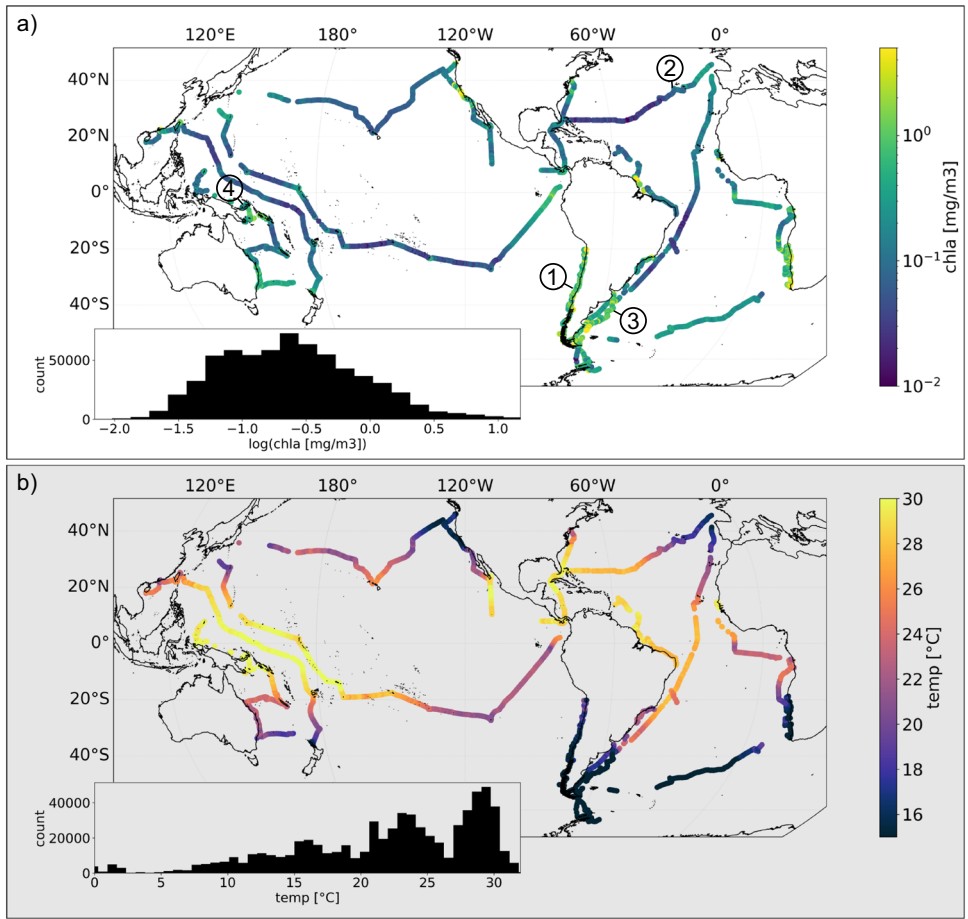

**Fig. 1 | The spatial extent and statistical distribution of the data used in this study.** Maps show the geographic distribution of chlorophyll-a (chla, **a**) and sea surface temperature (temp, **b**). The data includes 661,552 minute binned measurements. Insets show the histograms of both variables. The four numbers in panel (**a**) correspond to the example data shown in Fig. 2.

log-transformed, and we ran a least-squares fit to find the slope of this line, yielding the $\Gamma$ of $V = L^{\Gamma}$. In our analysis of global patterns of oceanic patchiness, we distinguish between $\Gamma$ of the biological and biogeochemical variables chl-a, $c_p(443)$ and $\gamma$ ($\Gamma_{\text{chl-a}}$, $\Gamma_{\text{cp}(443)}$ and $\Gamma_{\gamma}$, respectively), and of the physical variables temperature, salinity and density ($\Gamma_{\text{temperature}}$, $\Gamma_{\text{salinity}}$ and $\Gamma_{\text{density}}$, respectively).

Comparison between the geographic distributions of $\Gamma_{\text{chla}}$ and $\Gamma_{\text{temperature}}$ reveals distinct differences between global patterns of physical and biological patchiness (Fig. 3). Notably, the geographic distributions of $\Gamma_{\text{temperature}}$ are more random with lower autocorrelation compared to $\Gamma_{\text{chla}}$ at all spatial lags, while $\Gamma_{\text{chla}}$ is consistently lower in oligotrophic regions, particularly the subtropical gyres, and higher in physically energetic regions such as western boundary currents, eastern boundary upwelling zones, and equatorial upwelling zones. In agreement with most previous observations, $\Gamma_{\text{chla}}$ is, on average, lower than $\Gamma_{\text{temperature}}$, indicating chl-a is patchier than temperature, though there are exceptions, particularly in some coastal regions (Figure S4). $\Gamma_{\text{temperature}}$ is fairly normally distributed around 1, while $\Gamma_{\text{chla}}$ is slightly right-skewed with a mean around 0.5 (see inserts in Fig. 3). The -5/3 slope found in previous studies for the inertial subrange corresponds to a $\Gamma$ value of 0.84 which is near, but slightly lower than the mean of our global $\Gamma_{\text{temperature}}$ calculations, 0.97.

Interestingly, running the same analysis on satellite-based chl-a and temperature reveals contradicting results. Although satellite imagery has been successfully used to track individual chl-a patches[41,42], and to quantify patchiness at local scales[28], in the satellite-based analysis we find $\Gamma_{\text{temperature}}$ is lower than $\Gamma_{\text{chla}}$ (i.e. the opposite of our in situ

results), and the geographic patterns for both chl-a and temperature are different (Figures S5 and S6, respectively). We attribute this discrepancy primarily to issues of measurement sensitivity and atmospheric correction of ocean color remote sensing (from which chl-a is derived) which is applied as a pixel level correction, artificially injecting variance at small scales. This effect is likely to have the strongest impact in oligotrophic regions, where chlorophyll-a levels are particularly low, resulting in minimal absorption and reduced particulate scattering, and thus a low signal to noise ratio in the satellite retrieval. This could cause issues in both spatial and temporal analyzes[43]. This corroborates results from previous work that used a semivariogram approach to analyze variance in global satellite chl-a patterns from SeaWIFS[44] which showed that unresolved variance was negatively correlated with mean chl-a and the authors suggested the high fraction of unresolved variance in the subtropical gyres might stem from atmospheric correction, though they did not differentiate between instrument noise and submesoscale variability. A more recent, but similar study compared the unresolved variance between MODIS-Aqua and SeaWIFS and found it was reduced, but not eliminated with MODIS[45].

The global patterns we observe point to a difference between the governing processes underlying the formation of physical and biological ocean patchiness. This is emphasized when plotting the correlation between the different types of patchiness (Fig. 4), which we show both for all individual legs and averaged by Longhurst provinces, a partition of the global ocean into biogeochemical regions[46] (shown in Fig. 5). The $\Gamma$ of biological variables are correlated globally, even between chl-a and $\gamma$, the mean particle size (Fig. 4a, b). Similarly, $\Gamma$ of

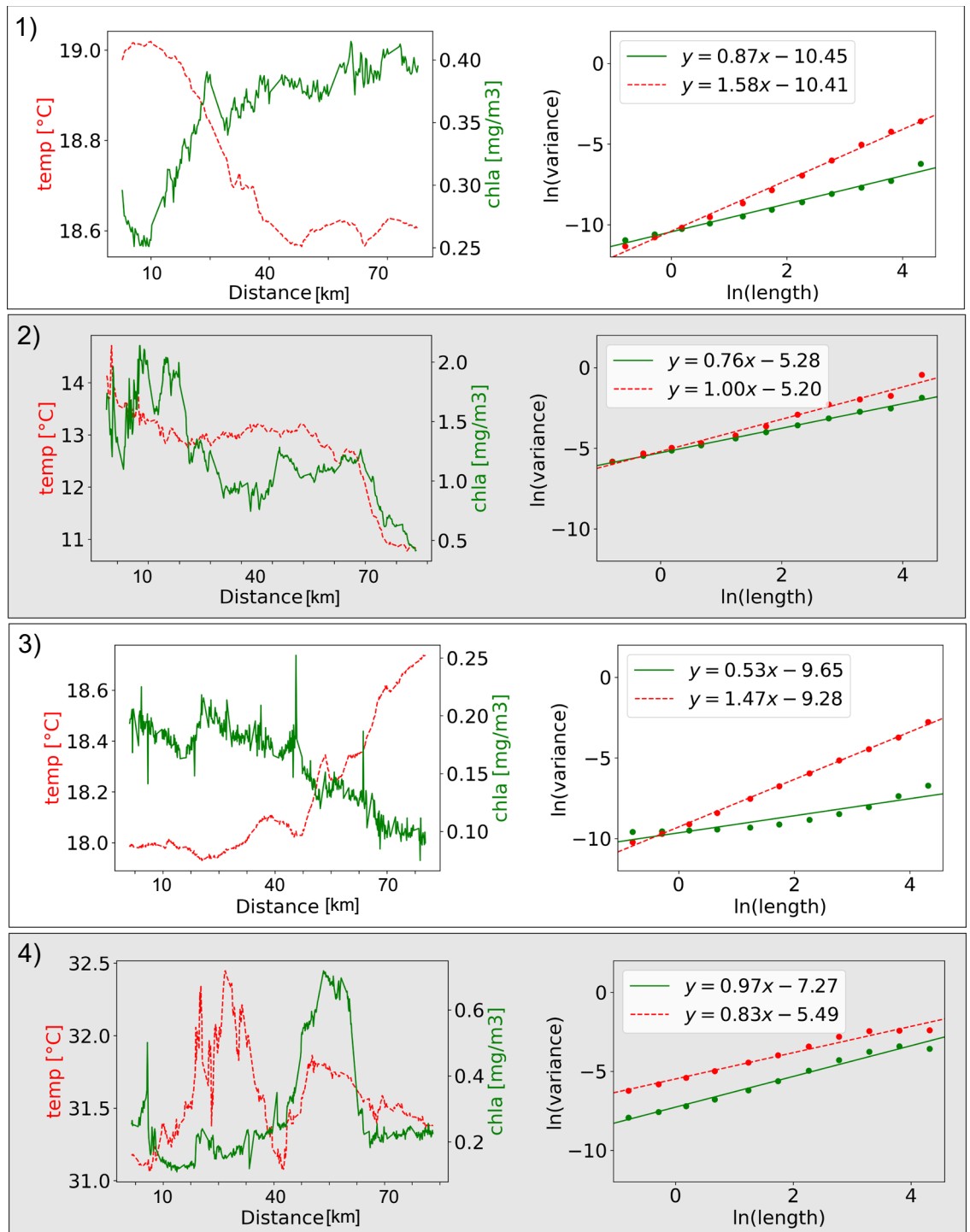

**Fig. 2 | Chlorophyll-a (chla) and temperature (temp) from four different transects and the resulting cascades of variance.** Examples of chla (green, solid) and temp (red, dashed) values along a leg (left panel in each pair) and the scale-dependent variance used to calculate the variance slope ($\Gamma$) derived from these legs (right panel in each pair). The variance slope can be seen in the legend for each panel on the right. The intercept indicated in each caption is a product of both the total amount of variance and the variance slope. The numbers of each panel (**1–4**) correspond to the locations in Fig. 1a.

physical variables are well correlated with each other (Fig. 4c, d). Yet, there is little to no correlation between the $\Gamma$ of biological variables and physical variables (Fig. 4e, f). This suggests that plankton patchiness is an emergent property due to biological processes that modulate the spatial patterns across scales sufficiently such that it is not correlated to the concurrent physical patchiness. Emergent is defined here as a property that emerges from the ensemble of processes in the ecosystem and environment.

If we decrease the top-end length scale that is used for the calculation of $\Gamma$, for example from 100 km to 5 km, $\Gamma_{chla}$ and $\Gamma_{temperature}$ are still uncorrelated ($R^2 = 0.04$ for 0.6 km to 5 km vs $R^2 = 0.02$ for 0.6 km to 100 km). This is in contrast with older work which showed that chl-a and temperature patchiness were correlated below 5 km yet not at scales from 5 km to 80 km[19].

Similarly to the differences between geographic distributions patterns (Figs. 1 and 3) the correlations between $\Gamma$ of different physical

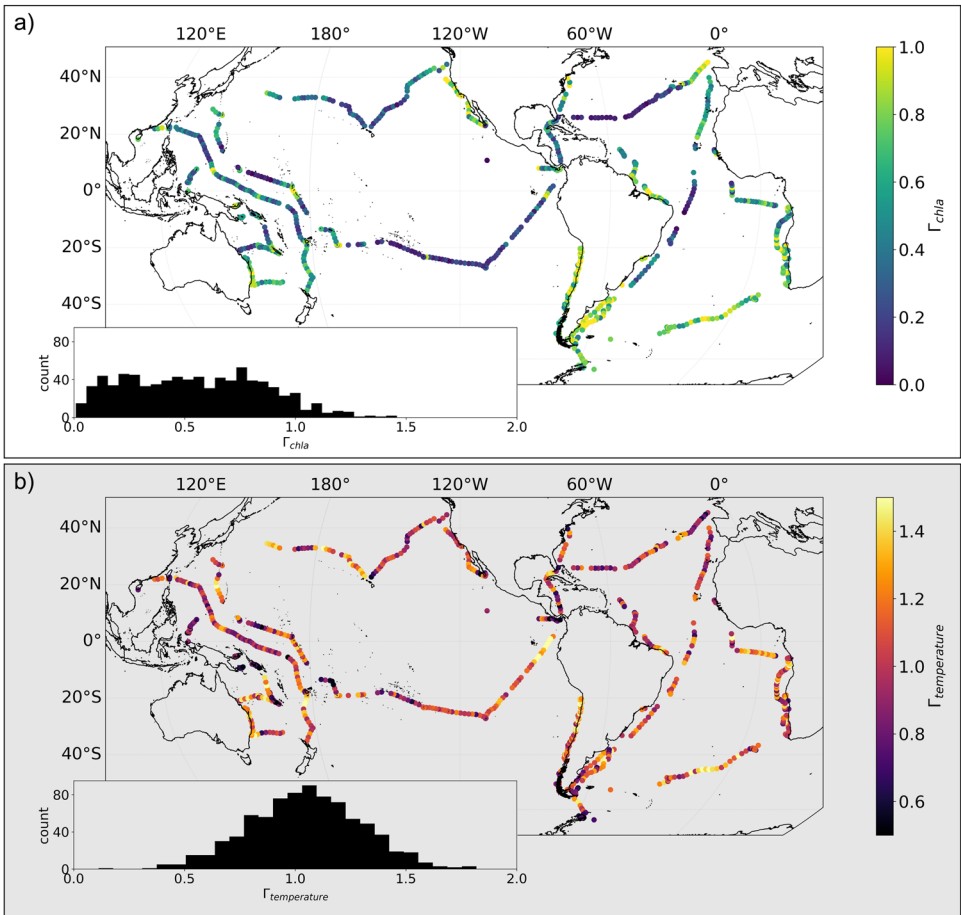

**Fig. 3 | Geographic distributions of variance slope ($\Gamma$).** Maps show the $\Gamma$ of chlorophyll-a ($\Gamma_{\text{chla}}$, **a**) and temperature ($\Gamma_{\text{temperature}}$, **b**). Insets in both panels show the histograms for these variables.

variables differ substantially from the correlations between absolute values (Figure S7). We show chl-a and $\gamma$ have minimal correlation in absolute value yet a correlation in $\Gamma$ ($R^2$ of 0.03 and 0.42 respectively), chl-a and temperature have a strong correlation in absolute value yet minimal in $\Gamma$ ($R^2$ of 0.36 and 0.02 respectively), and temperature and salinity have minimal correlation in absolute value yet a correlation in $\Gamma$ ($R^2$ of 0.04 and 0.25 respectively).

Another fundamental difference between patterns of physical and biological ocean patchiness emerges when looking at the data averaged by biogeochemical provinces and plotting the relationship between $\Gamma$ and the absolute value of each variable (Fig. 6). The biological variables that are concentration dependent (i.e. chl-a and $c_p$(443)) have a positive relationship between their $\Gamma$ and their absolute value, such that high biomass provinces are less patchy (i.e. associated with higher $\Gamma$) than low biomass provinces, (Fig. 6a–c). In contrast, there is no relationship between $\Gamma$ and the absolute value of the physical variables (Fig. 6d, e). In other words, biomass and biological patchiness are linked (inversely) while heat and salt content does not correlate with their patchiness.

The linkage between biological patchiness and biological productivity is emphasized when taking into account nutrient availability and oxygen saturation averaged by Longhurst Province (Fig. 7). When mapped onto the absolute value of chl-a and $\Gamma_{\text{chla}}$, a range of nutrients have a clear increasing relationship with both (Fig. 7), with the highest nutrient values also having high chl-a and $\Gamma_{\text{chla}}$. Oxygen saturation has a slight decreasing relationship with more considerable outliers.

The nature of the small-scale variability in biological signals, and the way they differ from the physical signals, is emphasized when

zooming in on segments of the transects (Fig. 8). Careful examination reveals that our underway observations of biological variables are characterized by high frequency changes that do not appear in the physical signals. While these high frequency changes may appear to be associated with instrumental noise, the individual measurements are robust, and the changes reflect small scale (< 1 km) biological variability patterns (Fig. 8). This robustness can be seen both in the covariation between different sensors (Fig. 8a) and the raw but consistent absorption and attenuation spectra (Fig. 8b, c). This small-scale variability lowers $\Gamma$ and can make the variance spectra nearly flat in regions without low frequency variation, such as the Sargasso Sea and Subtropical South Pacific.

Our work does not tie the variance slope to any particular process. Possibly $\Gamma$ represents the integrated impact of physical and biogeochemical characteristics within a region. The lower $\Gamma_{\text{chla}}$ in oligotrophic regions may be attributed to a sustained injection of variance from ecosystem interactions (e.g. growth, grazing). To investigate this line of thinking we quantified the patchiness of chromophoric dissolved organic matter (CDOM), which can be thought of as a more passive tracer than chl-a and a slow integrator of biological growth products. $\Gamma_{\text{CDOM}}$ appears to be intermediate with a small amount of variance explained by $\Gamma_{\text{chla}}$ and $\Gamma_{\text{temperature}}$ ($R^2$ of 0.17 and 0.06 respectively) and a statistical distribution closer to that of $\Gamma_{\text{temperature}}$ (Figure S8).

Recent work investigating zooplankton patchiness found coherent small-scale patches across many taxa dominated by length scales of 10-30 m[43] and even highly idealized models have been shown to generate patchiness via grazing[47] which could help explain aspects of our results. While previous work using this same metric suggested that

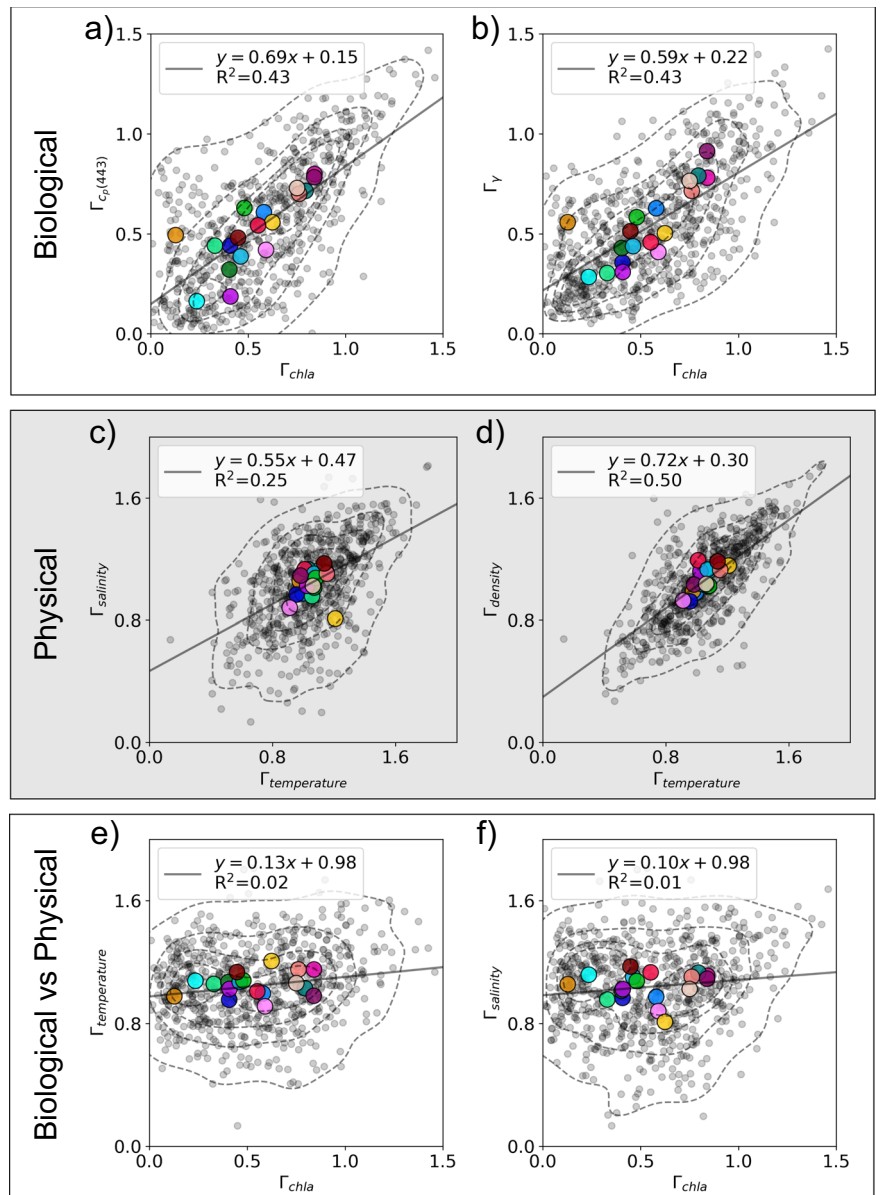

**Fig. 4 | Variance slope ($\Gamma$) comparisons between biogeochemical, physical, and biogeochemical vs physical variables.** $\Gamma_{chla}$ correlates with the $\Gamma$ of other biogeochemical parameters such as particulate attenuation ($\Gamma_{cp(443)}$, **a**) and mean particle size ($\Gamma_{\gamma}$, **b**), and $\Gamma_{temperature}$ correlates with the $\Gamma$ of other physical parameters such as salinity ($\Gamma_{salinity}$, **c**) and ($\Gamma_{density}$, **d**), yet the $\Gamma$ of biogeochemical vs physical variables don't have a notable correlation (**e**, **f**). Larger colored markers correspond with the Longhurst provinces from Fig. 5. Contours partition the data's probability mass function into five equal levels. N.b. correlations are shown for all data, not the Longhurst province means, and all plots have a *p*-value < 0.001.

the response time of phytoplankton (i.e. growth rate) is faster than the response time of temperature (i.e. equilibration time) as the reason chl-a is patchier[33], this doesn't agree with our results, where the regions with higher assumed growth rates are less patchy (e.g. eastern upwelling zones). This may suggest that certain elements of the ecosystem are responding faster in these highly patchy oligotrophic regions, possibly grazers or growth following fine-scale vertical nutrient fluxes. Stratification may play a role with a more stratified water column leading to more intermittent linkages between the surface and the nutrient-rich deep ocean, resulting in turn in a patchier expression in the highly stratified subtropical regions. While the drivers are not clear from our work, previous studies have noted that an increasingly patchy distribution requires finer scale sampling or grid spacing[33], and if these processes are important, our work indicates fine-scale grid spacing is in fact still important across broad swaths of the ocean generally considered "homogenous".

While the focus of this work is on scaling relationships, the fractal-like patchiness of phytoplankton is not infinite. At a fundamental level, patches are formed by individual phytoplankton living and dying[48]. Currently it is unclear how to connect the scales being investigated here with the individual organisms at the heart of the matter; Franks[25] notes that it is not likely we can diagnose the dynamics underlying the observations from spectral slopes alone. Towards mechanistic understanding of the processes underlying the spatial patterns we expect Lagrangian approaches will be key, particularly to parse out spatial and temporal dynamics. Future work should examine joint spatial patterning across trophic levels, combining the methods here, rapid methods for a wide size range of plankton[49], and methods for quantifying zooplankton patchiness[10]. Additionally, it may be that improved atmosphere correction of satellite products could enable a spatial analyzes across the global ocean. Higher signal to noise regions, such as coastal waters or

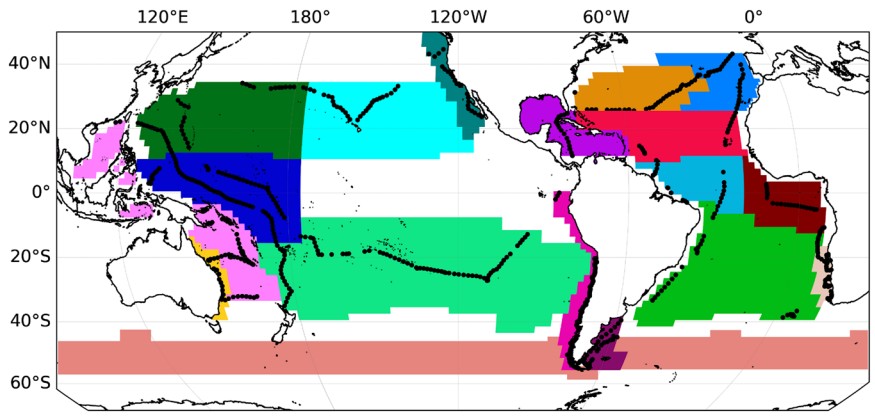

**Fig. 5 | Longhurst provinces with S/V _Tara_ legs overlaid in black.** Longhurst provinces that contain $N => 25$ legs are shown geographically along with the S/V _Tara_ legs within each province.

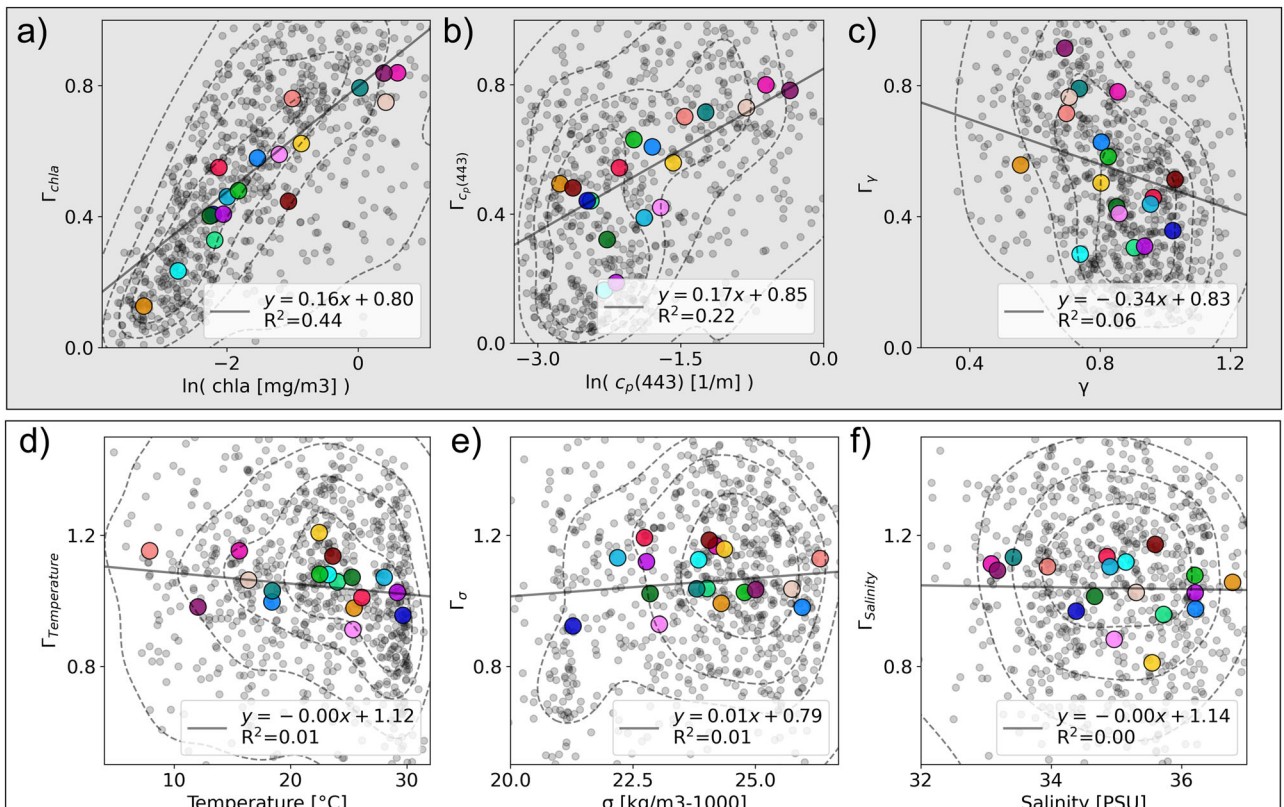

**Fig. 6 | Absolute value vs variance slope ($\Gamma$) for the variables under analysis.** The top rows (**a**–**c**) represent the biogeochemical variables: chlorophyll-a (chla), particulate attenuation at 443 nm ($c_p$(443)), and mean particle size ($\gamma$) and the and bottom rows (**d**–**f**) represent the physical variables: temperature, density ($\sigma$), and salinity. The average of each province is shown in the larger markers and the colors correspond to the provinces on the map in Fig. 5. Contours partition the data's probability mass function into five equal levels. N.b. correlations are shown for all data, not the Longhurst province means, and all plots have a _p_-value < 0.001.

higher latitudes during the spring bloom, may be more amenable to satellite analyzes of spatial patterns under existing atmospheric correction schemes.

We identified two fundamental differences between patterns of physical and biological patchiness: globally plankton patchiness at the scales analyzed here (~0.6 km–100 km) is spatially organized while physical patchiness is less coherent and organized differently, and patchiness of concentration-dependent biological variables is correlated with the absolute values of these variables, while no such correlation is found for physical variables. The statistical distributions of $\Gamma_{chla}$ and $\Gamma_{temperature}$ agree with previous work, where $\Gamma_{chla}$ has a

generally lower value, i.e. a patchier spatial pattern, compared to $\Gamma_{temperature}$. We find global patterns of biological patchiness to coincide with biogeochemical provinces and with nutrients levels. While our analysis shows the patchiness of biological variables are intercorrelated, and patchiness of physical variables are intercorrelated, contrary to common view, we find no correlation between physical and biological patchiness. This suggests that biological processes modulate the physically-driven advective baseline sufficiently to manifest as a different organization of patchiness. We emphasize here that the absolute value of variance itself is correlated at all scales between physical and biological variables (Figure S9). Moreover, while it is well

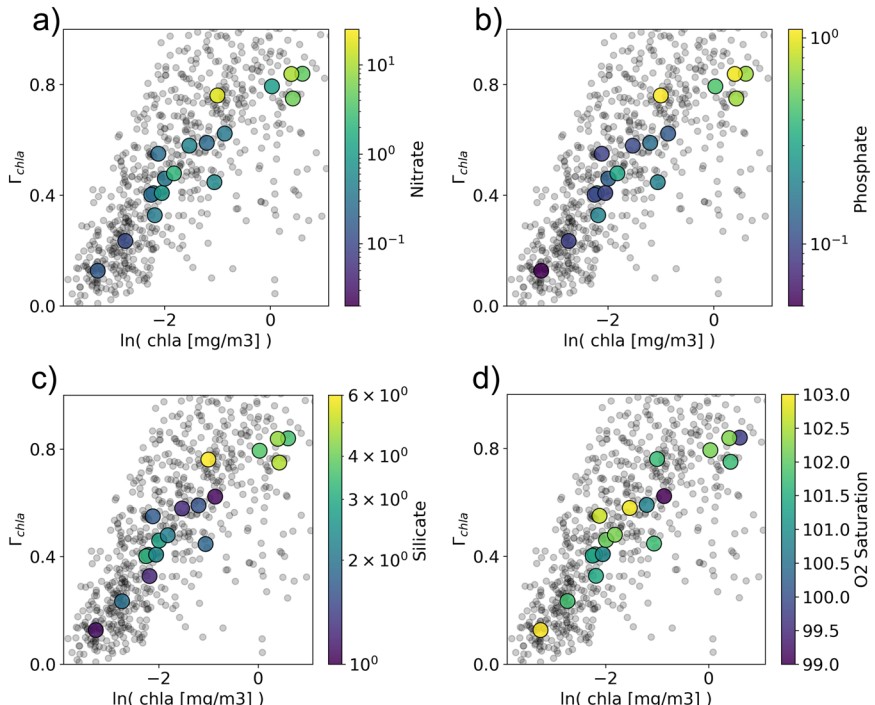

**Fig. 7 | Chlorophyll-a (chla) concentration vs variance slope of chla ($\Gamma_{chla}$) colored by biogeochemical parameters at the province level.** These are average nutrient concentration (**a**, **b**, **c**) and oxygen saturation (**d**) in each Longhurst province. Nutrient and oxygen saturation data is from the World Ocean Atlas 2018.

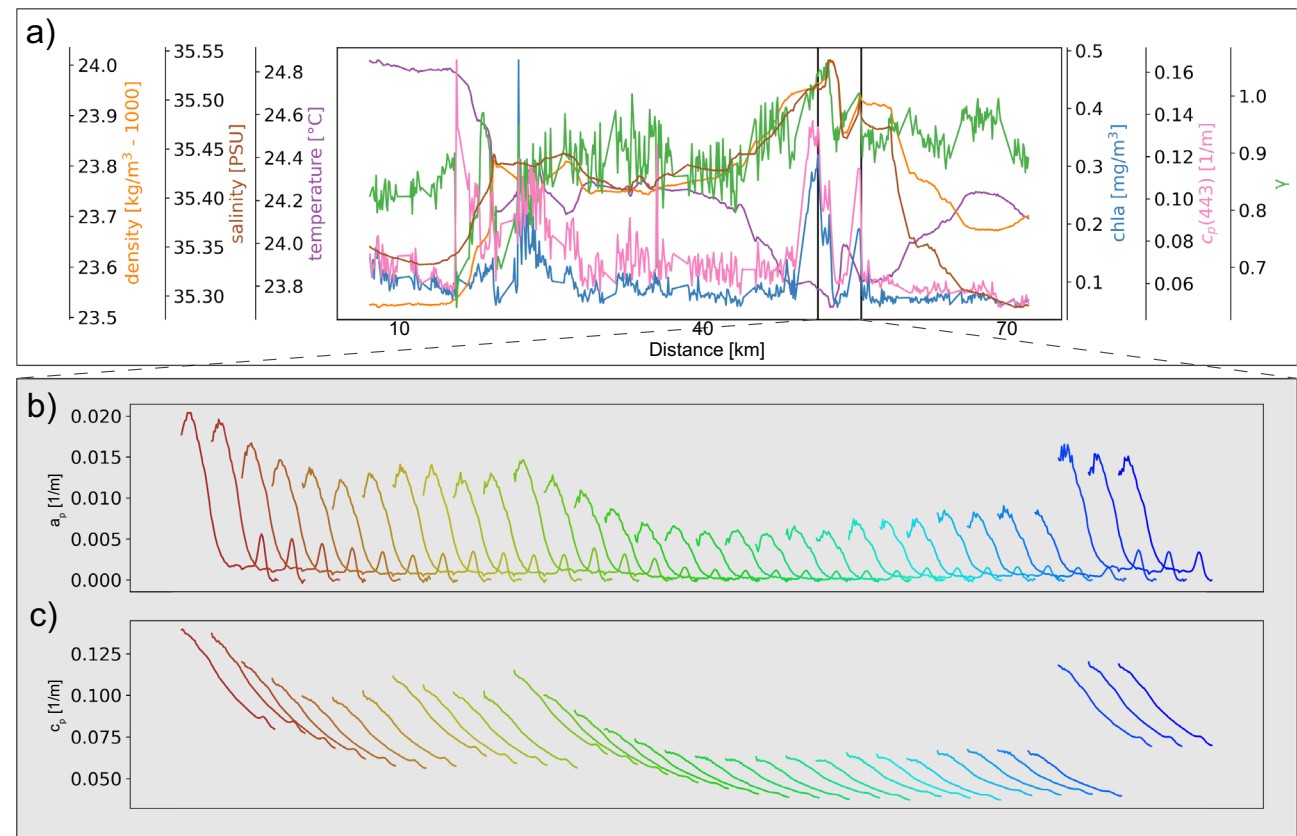

**Fig. 8 | High frequency environmental variability is evident in the raw data.** Here we show an example of variables along a single leg (**a**), showing data "spikes" that may initially be considered instrumental noise, but when inspecting the individual spectra from 408 nm to 730 nm (**b** and **c**), which are used to derive the various proxies, they are consistent with expectations for particulate absorption ($a_p$) and attenuation ($c_p$) data. The two bottom rows show a 30 minute subset of $a_p$ spectra (**b**) and $c_p$ spectra (**c**) representing the period between the vertical black lines in panel (**a**). Each spectra is the average of one minute of sampling, is colored by time (from brown to blue), and spans wavelengths from 410 nm to 750 nm. This transect is from the Coral Sea centered on 20.23 S, 152.90 E.

known that there is a relationship between temperature and chl-a, largely attributed to nutrient fluxes[50], our findings show this does not extend to variance slope. Our results contradict results from model studies (e.g[34].) and from satellite analyzes (e.g[51].), including our own (Figure S5 and S6). This discrepancy holds even when comparing to other work on the same range of scales as those examined here, which had concluded that biological patchiness is controlled by mesoscale mixing[28]. This suggests that these two important tools for understanding ocean ecosystems (models and satellite observations), fail to capture fundamental characteristics of the marine ecosystem. Our results may also help explain the disagreement shown across decades of patchiness research including many in situ studies. It may be that there is no generalizable patchiness relationship between physics and biology in the ocean and no consistent spatiotemporal scale where biological or physical processes dominate the spatial patterns of phytoplankton. Accordingly, basing general theories on the patchiness of a given tracer type from a specific site and over short time scales could be misguided. Instead, the ocean appears to consist of myriad physical and biogeochemical processes manifested as diverse spatial patterns. Thus, while no consistent relationships emerge these observable spatial patterns may provide a window into the processes creating them.

The results presented here support the use of low dimensional spatial encodings such as $\Gamma$ to study these upper-ocean processes. We speculate these spatial metrics may be sensitive to biogeochemical parameters not represented by the absolute value of chl-a. Importantly, we cannot resolve marine ecosystems at the mesoscale and below in basin-scale biogeochemical models, thus we must increase our ability to observe, invert, and understand what our models are missing[52]. Spatial patchiness patterns may help drive our models closer to reality and serve as an underutilized source of insight into the underlying processes generating the observed patchiness.

## Methods

All in situ measurements are from the S/V *Tara* underway sampling system which pulls seawater from 2 m depth. Temperature and salinity data is from a Seabird Scientific thermosalinograph (SBE 38 and SBE 45) and all optical data is derived from particulate absorption and attenuation measured with a Seabird Scientific ACs.

We quantify patchiness using variance slope, $\Gamma$, following[33] which is similar to spectral analysis, but able to ingest vectors (i.e. transects) with small gaps within them. $\Gamma$ is calculated on groups of 500 samples using a set of 11 log-distributed windows from 3 to 500. The median distance traveled between individual samples is 0.21 km (Figure S10) thus patchiness is calculated for scales from ~0.6 km to 100 km. Variance and window size were log-transformed, and we ran a least-squares fit to find the slope of this line, yielding the $\Gamma$ of $V = L^{\Gamma}$. It is worth noting a few characteristics of $\Gamma$ to gain some intuition into the metric. The variance slope of a randomly distributed variable is centered around 0. The calculation is not sensitive to whether a variable is log-normally distributed (such as chl-a). For example, $\Gamma_{chla}$ is identical whether raw chl-a values are input or log-transformed chl-a values. The S/V *Tara* is a sailing vessel and thus does not always travel in a straight line or with the same speed so here we have used time (minute bins of data) instead of distance to calculate the window, but filtered out legs that have a maximum distance traveled under 30 km or over 150 km. We have required at least 80% of the transect to contain data and filtered out transects where the maximum time elapsed is greater than 16 hours leading to a total of 820 transects used in this analysis.

Hyperspectral absorption and attenuation (400 to 735 nm at ~4 nm spectral resolution, AC-s, Seabird Sci.) were measured continuously. A 0.2 $\mu$m filter cartridge was connected to the system and we automatically redirected the flow to measure the properties of filtered seawater for 10 minutes every hour. Total ("normal") seawater was flowing the rest of the time. Absorption and attenuation measurements recorded during the filtered periods were interpolated across the transect and subtracted from the total seawater measurements to obtain an estimate of the particulate absorption ($a_p$), attenuation ($c_p$) spectra. During the Tara Microbiome transect, the filtered periods were interpolated using the variation in CDOM fluorescence (fCDOM) also measured continuously with a SeaPoint ultraviolet fluorometer (SUVF). fCDOM data was recorded with a Seabird Scientific WSCD CDOM fluorometer during the Tara Pacific transect. The signal to noise ratio of the WSCD is lower than the SUVF's which introduced noise in the fCDOM interpolation and the product derivation. Therefore, the filtered periods were interpolated linearly across the Tara Pacific transect. This approach permits retrieval of particulate optical properties independently from the instrument drift and biofouling[53]. For each minute of total seawater measurements (sampled at 4 Hz) the signal between the 2.5th and 97.5th percentiles were averaged, and their standard deviation was used to quantify uncertainty. Dropping the 2.5th to 97.5th percentiles filters out noisy spikes from bubbles which can be a major problem in optical measurements. All spectra were manually checked and quality controlled for obviously bad measurements (e.g. bubbles, bad filtered seawater measurements). These inherent optical properties were used as proxies for a range of particulate properties, primarily chl-a line height, a chl-a estimate derived from the absorption peak at 676 nm[54,55], and $\gamma$, a proxy for mean particle size[39].

Underway data was collected using Inlinino[56] an open-source logging and visualization program, and processed using InlineAnalysis (https://github.com/OceanOptics/InLineAnalysis) following best practices[57].

Longhurst provinces[46] (downloaded from https://www.marineregions.org/sources.php#longhurst) were used as approximate delineations of the global ocean into biogeochemical regions. Nutrient data was from the World Ocean Atlas 2018 (downloaded from https://www.ncei.noaa.gov/data/oceans/woa/WOA18).

### Reporting summary

Further information on research design is available in the Nature Portfolio Reporting Summary linked to this article.

## Data availability

All data is available in raw form at NASA's SeaBASS archive https://seabass.gsfc.nasa.gov/search via the search keyword "Tara_Microbiome". All data prepared and formatted for this study is available on GitHub https://github.com/patrickcgray/spatial_patchiness_tara, easily ingestible as a geofeather at multiple stages of processing.

## Code availability

All code necessary to generate the figures is on GitHub https://github.com/patrickcgray/spatial_patchiness_tara. To obtain a Docker image for running this in an identical environment to the one used in this study run "*docker pull pangeo/pangeo-notebook:2024.04.05*" from the command line with Docker Desktop running. A series of Jupyter Notebooks are provided in the above Github repo for an exact reproduction of all work in this study.

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

## Acknowledgements

We acknowledge support from the Zuckerman STEM Leadership Program to PCG. NSF EarthCube program award #2026932 supported the Pangeo Cloud platform where all analysis was conducted. The optical inline dataset was collected and analyzed with support from NASA Ocean Biology and Biogeochemistry program under grants NNX13AE58G, NNX15AC08G and 80NSSC21K0783, and 80NSSC20K1641 to the University of Maine. We wish to thank the Tara Ocean Foundation, the SV Tara crew and all those who participate in Mission Microbiomes AtlantECO and adopt its Data Sharing & Publication Best Practices (https://zenodo.org/communities/mission-microbiomes-atlanteco/). This publication has received funding from the European Union's Horizon 2020 research and innovation program under grant agreement No 862923 (project AtlantECO). This output reflects only the author's view and the European Union cannot be held responsible for any use that may be made of the information contained therein. We are keen to thank the commitment of the following institutions for their financial and scientific support that made Mission Microbiomes AtlantECO possible: Stazione Zoologica Anton Dohrn, European Bioinformatics Institute (EMBL-EBI), Center national de la recherche scientifique (CNRS), Center National de Séquençage (CNS, Genoscope), agnès b., BIC, Capgemini Engineering, Fondation Groupe EDF, Compagnie Nationale du Rhône, L'Oréal, Biotherm, Région Bretagne, Lorient Agglomération, Billerudkorsnas, Havas Paris, Fondation Rothschild, Office Français de la Biodiversité, AmerisourceBergen, Philgood Foundation, UNESCO-IOC, Etienne Bourgois. Special thanks to the Tara Ocean Foundation, the S/V Tara crew and the Tara Pacific Expedition Participants (https://doi.org/10.5281/zenodo.3777760). We are keen to thank the commitment of the following institutions for their financial and scientific support that made this unique Tara Pacific Expedition possible: CNRS, PSL, CSM, EPHE, Genoscope, CEA, Inserm, Université Côte d'Azur, ANR, agnès b., UNESCO-IOC, the Veolia Foundation, the Prince Albert II de Monaco Foundation, Région Bretagne, Billerudkorsnas, AmerisourceBergen Company, Lorient Agglomération, Oceans by Disney, L'Oréal, Biotherm, France Collectivités, Fonds Français pour l'Environnement Mondial (FFEM), Etienne Bourgois, and the Tara Ocean Foundation teams. Tara Pacific would not exist without the continuous support of the participating institutes. The authors also particularly thank Serge Planes, Denis Allemand, and the Tara Pacific consortium. Mission Microbiomes CEODOS Chile expedition would not exist without the leadership of the Tara Ocean Foundation and the continuous support of many research institutes. Special thanks to the Tara Ocean Foundation teams, the R/V Tara crew and the Mission Microbiomes CEODOS Chile Expedition Participants. We are keen to thank the commitment of the following institutions for their financial and scientific support that made this unique Mission Microbiomes CEODOS Chile Expedition possible: COPAS Coastal Center, Center for Mathematical Modeling (CMM), Center for Dynamic Research of High Latitude Marine Ecosystems (IDEAL), Patagonia Ecosystem Research Center (CIEP), Interdisciplinary Center for Aquaculture Research (INCAR), Center for Climate and Resilience (CR2), University of Chile, University of Concepción, Universidad Austral de Chile, CNRS (in particular the Research Federation for the Study of Global Ocean Systems Ecology and Evolution FR2022/Tara Oceans-GOSEE), CEA-Genoscope, Ministry of Science, Technology, Knowledge, and Innovation of Chile, Chilean Navy and SHOA. We also thank agnès b. and Etienne Bourgois, the Veolia Foundation, Region Bretagne, Lorient Agglomeration, Compagnie Nationale du Rhone, BIC, Biotherm, L'OREAL, ALTRAN, Office Français de la Biodiversité (OFB), Fonds Français pour l'Environnement Mondial (FFEM), Billerudkorsnas, UNESCO-IOC, the Prince Albert II de Monaco Foundation, AmerisourceBergen Company, Oceans by Disney and France Collectivités, for their sponsorships and commitment. The global sampling effort was enabled by countless scientists and crew who sampled aboard the Tara schooner in 2021. We are also grateful to the countries who graciously granted sampling permission. The authors declare that all data reported herein are fully and freely available from the date of publication, with no restrictions, and that all the analyses, publications, and ownership of data are free from legal entanglement or restriction, except commercialization bilateral agreements, by the various nations whose waters the Mission Microbiomes CEODOS Chile expedition sampled in. The consortium also benefited from the particular grants: France Génomique (ANR-10-INBS-09), COPAS COASTAL ANID FB210021, Center for Mathematical Modeling (CMM) BASAL fund FB210005 for center of excellence from ANID-Chile, Millennium Institute Center for Genome Regulation (Project ANID–MILENIO-ICN2021_044), POGO CEODOS Chile working group, IRP MAST CNRS. This article is contribution number 1 of Mission Microbiomes CEODOS Chile consortium.

## Author contributions

The study was conceived by all authors. The analysis approach was architected by Y.L. The specific methods and analysis were conducted by P.G. Data collection was led by E.B and G.B. Data processing was led by G.B. All authors interpreted the results, wrote the paper, and supported data analysis. The Mission Microbiomes AtlantECO and Tara Pacific Consortiums supported the extensive collection of the data and logistics.

## Competing interests

The authors declare no competing interests.

## Additional information

## Mission Microbiomes AtlantECO

A. Bourdais[3], C. Bowler[4], C. Moulin[3], C. de Vargas[5], D. Iudicone[6], D. Couet[5], E. Catafort[7], E. Boss[1], E. Petit[8], E. Mayeux[8], F. Lombard[5], J. Schramm[3], L. Guidi[5], M. Moll[9], P. Wincker[8], R. Laxenaire[10], R. Troublé[3], S. Sanchez[8], S. Pesant[11] & T. Linkowski[3]

[3]Tara Ocean Foundation, Paris, France. [4]Centre national de la recherche scientifique, Paris, France. [5]Sorbonne Université, Paris, France. [6]Stazione Zoologica Anton Dohrn, Naples, Italy. [7]World Courier, Paris, France. [8]Centre de l'énergie atomique, Paris, France. [9]EMS Sistemas, Barcelona, Spain. [10]Ecole Normale Supérieure, Paris, France. [11]European Molecular Biology Laboratory, Heidelberg, Germany.

## Tara Pacific Consortium

S. Planes[12], D. Allemand[13], N. Djerbi[14], B. C. C. Hume[15], T. Röthig[16], M. Ziegler[17], L. Paoli[18], J. M. Flores[19], N. Lang-Yona[20], P. Conan[21], P. E. Galand[22], E. Douville[23], S. Agostini[24], Y. Kitano[25], O. da Silva[26], D. R. Cronin[27], E. Armstrong[28], J. -M Aury[28], B. Banaig[12], Barbe V[28], C. Belser[28], E. Beraud[13], E. Boissin[12], G. Klinges[29], E. Bonnival[30], E. Boss[1], G. Bourdin[1], E. Bourgois[3], C. Bowler[4], Q. Carradec[28], S. Pesant[31], M. Miguel-Gordo[32], N. Cassar[33,34], S. G. John[35], N. R. Cohen[36], G. Reverdin[37], J. Filée[38], C. de Vargas[5], J. R. Dolan[26], G. Dominguez Herta[27], J. Du[27], D. Forcioli[14], R. Friedrich[7], P. Furla[14], J. -F Ghiglione[21], E. Gilson[14], G. Gorsky[26], M. Guinther[24], N. Haëntjens[1], N. Henry[30], M. Hertau[3], C. Hochart[22], G. Iwankow[12], L. Karp-Boss[1], R. L. Kelly[35], I. Koren[19], K. Labadie[28], J. Lancelot[3], J. Lê-Hoang[28], R. Lemee[26], Y. Lin[33], F. Lombard[26], D. Marie[30], R. McMind[14], M. Trainic[19], D. Monmarche[3], C. Moulin[3], Y. Mucherie[3], B. Noel[28], A. Ottaviani[14], M. -L Pedrotti[26], C. Pogoreutz[15], J. Poulain[28], M. Pujo-Pay[21], S. Reynaud[13], S. Romac[30], E. Rottinger[14], A. Rouan[14], H. -J Ruscheweyh[18], G. Salazar[18], M. B. Sullivan[27], S. Sunagawa[18], O. P. Thomas[32], R. Troublé[3], A. Vardi[20], R. Vega-Thunder[29], C. R. Voolstra[15], P. Wincker[28], A. Zahed[27], T. Zamoum[14] & D. Zoccola[13]

[12]PSL Research University: EPHE-UPVD-CNRS, USR 3278 CRIOBE, Université de Perpignan, Perpignan, France. [13]Centre Scientifique de Monaco, Principality of Monaco, Monaco, Monaco. [14]Université Côte d'Azur, CNRS, Inserm–IRCAN, Nice, France. [15]Department of Biology, University of Konstanz, Konstanz, Germany. [16]Aquatic Research Facility, Environmental Sustainability Research Centre, University of Derby, Derby, United Kingdom. [17]Department of Animal Ecology & Systematics, Justus Liebig University, Giessen, Germany. [18]Department of Biology, Institute of Microbiology and Swiss Institute of Bioinformatics, ETH Zürich, Zürich, Switzerland. [19]Weizmann Institute of Science, Dept. Earth and Planetary Science, Rehovot, Israel. [20]Weizmann Institute of Science, Dept. Plant and Environmental Science, Rehovot, Israel. [21]Sorbonne Université, CNRS, LOMIC, Observatoire Océanologique de Banyuls, Banyuls-sur-Mer, France. [22]Sorbonne Université, CNRS, LECOB, Observatoire Océanologique de Banyuls, Banyuls-sur-Mer, France. [23]Laboratoire des Sciences du Climat et de l'Environnement, LSCE/IPSL, CEA-CNRS-UVSQ, Université Paris-Saclay, Gif-sur-Yvette, France. [24]Shimoda Marine Research Center, University of Tsukuba, Shizuoka, Japan. [25]National Institute of Environmental Science, Tsukuba, Japan. [26]Sorbonne Université, Institut de la Mer de Villefranche sur mer, Laboratoire d'Océanographie de Villefranche, Villefranche-sur-Mer, France. [27]The Ohio State University, Departments of Microbiology and Civil, Environmental and Geodetic Engineering, Columbus, Ohio, United States of America and The Ohio State University, Departments, Columbus, Ohio, United States of America. [28]Génomique Métabolique, Genoscope, Institut François Jacob, CEA, CNRS, Univ Evry, Université Paris-Saclay, Evry, France. [29]Oregon State University, Department of Microbiology, Corvallis, Oregon, United States of America. [30]Sorbonne Université, CNRS, Station Biologique de Roscoff, AD2M, UMR 7144, ECOMAP, Roscoff, France. [31]PANGAEA, Data Publisher for Earth and Environment Science, Bremen, Germany & MARUM—Center for Marine Environmental Sciences, Universität Bremen, Bremen, Germany. [32]Marine Biodiscovery Laboratory, School of Chemistry and Ryan Institute, National University of, Ireland Galway, Ireland. [33]Division of Earth and Ocean Sciences, Duke University, Durham, North Carolina, United States of America. [34]Laboratoire des Sciences de l'Environnement Marin LEMAR., UMR 6539 UBO/CNRS/IRD/IFREMER, Institut Universitaire Européen de la Mer IUEM, Brest, France. [35]Department

of Earth Sciences, University of Southern California, Los Angeles, California, United States of America. [36]Marine Chemistry and Geochemistry Department, Woods Hole Oceanographic Institution, Falmouth, Massachussettes, United States of America. [37]Institut Pierre Simon Laplace, CNRS/IRD/MNHN LOCEAN. Sorbonne-Université Paris, Paris, France. [38]Laboratoire Evolution, Génomes, Comportement et Ecologie, CNRS/Université Paris-Saclay, Avenue de la Terrasse, Gif sur Yvette, France.

## Members of the Mission Microbiomes CEODOS Chile consortium

**Camila Fernández[39,40,41], Alejandro Maass[42,43,44,45], Chris Bowler[4], Leonardo Castro[39,40], Giovanni Daneri[39,46], Colomban De Vargas[5], Damien Eveillard[45,47], Laura Farías[40,48,49], Humberto González[50,51], Lionel Guidi[5], Fabien Lombard[26], Daniele Iudicone[6], Silvio Pantoja[39,40], Renato Quiñones[40,52], Romain Trouble[3] & Patrick Wincker[8]**

[39]Center for Oceanographic Research in the Eastern South Pacific COPAS COASTAL, Concepción, Chile. [40]Department of Oceanography, University of Concepción, Concepción, Chile. [41]Laboratory of Microbial Oceanography, UMR7621, Banyuls sur Mer CNRS, Banyuls-sur-Mer, France. [42]Department of Mathematical Engineering, University of Chile, Santiago, Chile. [43]Center for Mathematical Modeling, University of Chile and IRL2807 CNRS, Santiago, Chile. [44]Millennium Institute Center for Genome Regulation, Santiago, Chile. [45]Research Federation for the Study of Global Ocean Systems Ecology and Evolution, FR2022/Tara Oceans GOSEE, Paris, France. [46]Centro de Investigación en Ecosistemas de la Patagonia (CIEP), Coyhaique, Chile. [47]Nantes Université, École Centrale Nantes, CNRS, LS2N, UMR 6004, Nantes, France. [48]Instituto Milenio en Socio-Ecología Costera (SECOS), Santiago, Chile. [49]Center for Climate and Resilience Research (CR2), University of Chile, Santiago, Chile. [50]Centro FONDAP de Investigación en Dinámica de Ecosistemas Marinos de Altas Latitudes (IDEAL), Punta Arenas, Chile. [51]Instituto de Ciencias Marinas y Limnológicas, Universidad Austral de Chile, Valdivia, Chile. [52]Interdisciplinary Center for Aquaculture Research (INCAR), Universidad de Concepción, Concepción, Chile.

