## [Transparent Peer Review file · Nature Communications]

Emergent patterns of patchiness differ between physical and planktonic properties in the ocean

Corresponding Author: Dr Patrick Gray

Version 0:

Reviewer comments:

Reviewer #1

(Remarks to the Author)

Summary: The authors perform a global analysis to show that some biological and physical parameters are decorrelated across spatial scales. Although this analysis is fairly comprehensive and relies on extensive data, I have several concerns with the data interpretation and the robustness of the results. With some extensive modification, I could see this manuscript being a useful and timely contribution.

Major concerns:

1. The terms 'ocean biology' and 'ocean physics' are far too broad for the data that are actually being studied. There are many components of ocean biology, such as phytoplankton diversity, physiological traits, etc. that are not directly captured in this study. Similarly, the influence of ocean physics goes beyond temperature and salinity (wind speed, ocean currents etc.). I would suggest that the authors limit their title, abstract and major conclusions to prevent over-interpretation of the data actually being studied. In the best-case scenario, this study is arguing for a decorrelation (not decoupling) between phytoplankton biomass and seawater density, not ocean physics and biology.

2. Although I understand the reasons for using linear regression to identify the strength of the relationship, the distribution of points for many of the plots shows that the fit is not optimal. The effect sizes are also low. In Fig. 4, the greatest R² value is 0.46 (between Γ_{density} and Γ_{temp}), which makes sense, given the known relationship between seawater density and temperature. Otherwise, there are no additional analyses to support the arguments throughout the paper. In its current form, I am not entirely convinced that the data support the authors' conclusions (Ln 226-235). Here are some suggestions:

- It might be interesting to highlight where Γ_{chl} and Γ_{temp} are equal, where Γ_{chl} exceeds Γ_{temp} and where Γ_{chl} is less than Γ_{temp} on a map. This could add nuance to your arguments and provide more context for the results.
- Consider reinforcing the correlation plots with supplementary figures showing what the residuals look like.
- Density plots could indicate where the points are concentrated when comparing any 2 variables. The Longhurst provinces all seem to cluster around the middle, with some interesting exceptions.

3. Ln 261-270: All of Fig.6 just seems to show clouds of points with no obvious correlation. The regression lines also seem weird (6a-c) compared to the reported equations.

- Why are the intercepts so different in the equations (~0.7)? The lines on the plots show an intercept less than 0.6.
- In 6a, why does Γ_{chl} only go up to 0.9?
- In 6c, why does the line on the actual plot appear to show positive slope when it is -0.21?

These are some major irregularities.

4. Ln 279-283: O₂ saturation is not a nutrient. In this case as well, I am not sure why Γ_{chl} only goes up to 0.9 when Fig. 2 shows the histogram going up to 1.5.

5. Some additional thoughts and questions:

- From what I understand, the data presented here were all measured right at the surface. It might be useful to discuss whether vertical mixing rates, phytoplankton mobility, buoyancy (species composition) could alter observed patchiness on short time scales.

- I am surprised there was no mention of local wind patterns on observed chlorophyll patchiness, especially given that the sampling was conducted on a sailing vessel (Ln 413). Instead of temperature and salinity, my understanding was that wind, currents and mixed layer depth tend to be important in determining patchiness.

George, D. G., & Edwards, R. W. (1976). The Effect of Wind on the Distribution of Chlorophyll A and Crustacean Plankton in a Shallow Eutrophic Reservoir. *Journal of Applied Ecology*, 13(3), 667–690. <https://doi.org/10.2307/2402246>

Kahru, M., S. T. Gille, R. Murtugudde, P. G. Strutton, M. Manzano-Sarabia, H. Wang, and B. G. Mitchell (2010), Global correlations between winds and ocean chlorophyll, *J. Geophys. Res.*, 115, C12040, doi:10.1029/2010JC006500.

Overall, by generalizing over the entire dataset and not really looking into the actual spatial patterns, the conclusion that there is “a decoupling between ocean physics and biology” seems overstated.

Additional comments:

Fig. 2 (right panel): Might be useful to indicate the Γ values more explicitly for readers. Showing the entire equation is fine, but I expected some discussion on the intercept values (the fourth panel looks different). It is also interesting to me where the red and green lines cross (at what length scale). Also, consider using a color-blind friendly combination.

Fig. 3: Why is the range of the distribution greater than the legend on the right? It seems the values for Γ_{chl} go up to 1.5 and Γ_{temp} go up to 1.9, but the legend ends at 1 and 1.5.

Ln 239: “Marginally more correlated” is quite the bold claim. When comparing an R^2 of 0.04 vs 0.03, I think it is safe to say that Γ_{chl} and Γ_{temp} are not correlated in both cases.

Fig. S6: Please check the equations for the last 2 panels.

Reviewer #2

(Remarks to the Author)
Review Gray et al.

This paper examines a large data set of underway, near-surface, bio-optical, temperature and salinity data from the S/V Tara. The data set and the information it holds is impressive, as the ship's track covers the Atlantic and Pacific, with multiple basin crossings, as well as tracks along the basin margins. The authors examine the patchiness of physical (temperature and salinity) and biogeochemical (chlorophyll-a, beam attenuation $cp(443)$ and γ .) variables using the exponent of the variance vs length-scale (V-L) relationship for each of the variables. The exponent can be used as a patchiness measure, as shown previously in ref. 31.

Consistent with previous literature, the authors find that the patchiness (exponent of the V-L relationship) differs between physical and biogeochemical variables. They also find that the patchiness of physical variables is not correlated with the patchiness of biological variables across different regions of the oceans. This leads them to conclude that “patchiness of physics and biology are decoupled at the global scale.” This conclusion is misleading. On first reading, it would be easily mis-understood that the authors conclude that patchiness of biological variables is not related to the patchiness of physical variables. This would be wrong. The physics does indeed affect the patchiness of biology. I think that the authors' conclusions are either incorrect or mis-stated.

Assuming that the conclusions (interpretation) is mis-stated — it would be worth understanding why the authors expect the exponent of physical and biological variables to be correlated. Even if the distribution of biological variables were shaped by the physics, the difference in the exponent of the V-L relationship could vary depending on the time scales of the biological processes as compared to the physical processes. Their ratio could differ regionally and hence the findings should not lead to the conclusion that the authors have drawn.

There are several other issues (inaccuracies) in this paper. The term “global scale” is used incorrectly. The authors mean to convey that the relationship between the regional-scale patchiness of temperature and chlorophyll-a differs from one region to another. This result is not surprising nor does it challenge any of the previous literature that states that the patchiness of biogeochemical and physical variables is related. The result does not pertain to the global scale (largest scales).

Please see: Campbell, J.W., Lognormal distribution as a model for bio-optical variability in the sea, JGR 1995, which addresses some of the questions raised in the manuscript.

What exactly is “gamma”? The paper refers to this only once and explains it as an optical proxy for mean particle size.

Methods: In the text, the authors say the spacing between measurements is about 150m. But since the measurements are at certain time intervals, the lateral spacing between 2 measurements could differ considerably. The authors do the analysis using windows defined by time (lines 414-415). This could affect the results significantly. It is just like calculating spectra for unevenly spaced data, assuming equal spacing. At the least, the authors need to show the spacing - and the variability in the spacing of measurements.

Supplementary materials was not very helpful because the figure captions were inadequate and there was no supporting text.

Line 23: slope of Variance vs. Spatial scale (rather than spatial scale vs. variance)

Line 27: Incorrect to state that the patchiness of physical and biological parameters are uncorrelated.

Line 28 What is meant by “variance slope is an emergent property”?

Line 30 No, this does not “provide context for decades of discrepancy”

Line 31 New tests? What is meant by this?

Line 32 insight instead of “new insight”

Line 40 that result in

Line 43 “with close proximity” What does this mean?

Line 49 Is there a reference to substantiate this statement? “generally increasing patchiness as one moves up the trophic chain”

Line 72 “this parameter” — Spectral slope?

Line 73 replace “thought of as quantifying” by “interpreted”

Line 75 It is not a decomposition

Lines 78-90 —There are many incorrect or inaccurate statements in this section.

Line 79 Energy cascades, not turbulence

Line 79 Replace “turbulent mesoscale” by “mesoscale” as the large scale turbulence is different from 3D turbulence

Line 81 No, this is not how it works. Energy is transferred up scale at small Rossby number (the downscale cascade is facilitated by submesoscale processes)

Lines 84-87 This is confusing and misleading. The inertial subrange of Kolmogorov is different from the large-scale dynamics of the ocean - which is affected by rotation and stratification.

Line 88-90 Temperature could have a variance source at smaller scales too — for example, due to upwelling on small scales.

Line 92 Previous literature did not attribute phytoplankton patchiness simply to turbulent stirring

Line 103 These references (7,30,31) did not all corroborate each other — on the contrary, they gave different reasons for the observed patterns in patchiness of temperature and chlorophyll.

Line 105 What is the “new” heat flux?

Line 108 The SST equilibrates with the atmosphere largely due to air-sea heat exchanger and the statement “heat is mixed over time” is not correct.

Line 109 “return to covarying” - not correct

Lines 103-113 The references are inaccurate

Line 115 What is meant by “actual observations remain inconsistent” ?

Lines 116-119 - could use a better word than “work” which appears repeatedly

Line 120 “yet not so above 5 km to 80 km” what does this mean?

Line 142 Measurements of what?

Line 173 (fig 2 caption) What are “absolute” values

Line 194 — related to the reference by Campbell (see above)

Line 206-208 - previous papers have not reported contradictory results — not sure why the authors find contradictory results when using satellite data

Line 214-216 Could this be because chlorophyll is log normally distributed

Line 234-235 - No. This conclusion is not substantiated by the findings.

Line 273

Fig 7. Same as Fig 6a except for the color of dots?

Fig 8. Could this figure go in the supplementary material?

Line 315 scale

Line 320 - seems that Reference 31 is for oligotrophic regions, whereas the results presented here are for productive regions

Line 353 - “physical patchiness is more random” - this does not seem correct

Line 361-2 why should there be an interdependence?

Line 365 - this is not a conclusion that can be drawn from the results presented - and seems incorrect

Due to the incorrect interpretation of findings, several inaccuracies, and understanding gaps of previous literature, I do not recommend publication of this article.

Reviewer #3

(Remarks to the Author)

This article examines the scales of variability in both optical and physical variables from in-situ ocean surface observations. Results demonstrate that basin-scale variance slope does not correlate between the biological/biogeochemical and physical variables. A shorter analysis of similar variables from satellite data (mainly in the supplement) demonstrates one reason why previous studies have found correlations in spectral slopes or variance slopes, as those do occur in the satellite record. These results are critical for understanding decades of disagreements in the field, as it seems there will not be a simple relationship to be found between the variance slope for physical and biological variables, but rather only an underlying set of relationships between the physical environment and biological concentrations, and then from concentrations to variance slope. This is an important step toward improving biogeochemical and climate models and satellite data analysis in the future.

This is a well-written article with clear methods and results. The analysis supports the conclusions drawn. There are no major flaws. I recommend this article be returned for minor revisions to address comments below.

Minor Revisions

1. Line 120, citation 17. Is the correlation below 5km and maybe above 80km, or is the correlation occurring at small scales with an uncertain upper limit between 5 and 80 km? Please rewrite for clarity.
2. At the beginning of the results, please repeat or move (from line 198) the equivalent value for Gamma (variance slope) to $-5/3$ for spectral slope. I really wanted to know at that time when reading.
3. In Figures 1 and 3, the dots on the map are quite small. Is it possible to increase their size while still allowing the gaps to be visible? Maybe they can be drawn as ellipses or rectangles so that there are larger colored areas without excessive overlap.
4. Line 230: please add a brief description of Longhurst provinces and note that Figure 5 shows their location-- I read this as the averages would be in Figure 5 separately from the full data in Figure 4.
5. Figure 6c: Please add a note as to why the x-axis is reversed or remake this with an increasing x-axis. It is quite jarring to see a negative slope in the labeled equation with the visually 'positive' slope. Also please explain why $R^2=0.02$ is considered correlated and $R^2=0.00$ is not. Even though p is small, it is hard to understand why this is convincing. Would showing the log be better?
6. Figure 7d: The Longhurst province averages for O2 do not appear to be correlated to chl-a or Gamma_chla here. Please confirm that this is the case, as you assert (line 282) that this is a clear increasing relationship and by eye it is not.

Typos Etc.

1. Line 290: add 'in' after 'zooming'
2. Figure 8: please label the location this data is from in more detail-- can you give a latitude/longitude for the middle, perhaps?
3. The end of the results section includes author names for references, which is not the case elsewhere. Please double check these choices.
4. In the references, sometimes 'Boss' does not have initials for first/middle name.
5. Please check references to make a consistent format. The number of authors included, for example, before going to et al., is variable.

Version 1:

Reviewer comments:

Reviewer #1

(Remarks to the Author)

I think the revised text and figures more closely align with the stated claims in the manuscript. My only suggestion would be to perform a careful proofreading, as I noticed some minor grammatical errors and miscitations.

Reviewer #3

(Remarks to the Author)

This is a well-written article with clear methods and results. The analysis supports the conclusions drawn. There are no major flaws. The revised manuscript has addressed all my previous minor concerns. I recommend this article be published in this journal.

Broader description:

This article examines the scales of variability in both optical and physical variables from in-situ ocean surface observations. Results demonstrate that basin-scale variance slope does not correlate between the biological/biogeochemical and physical variables. A shorter analysis of similar variables from satellite data (mainly in the supplement) demonstrates one reason why previous studies have found correlations in spectral slopes or variance slopes, as those do occur in the satellite record. These results are critical for understanding decades of disagreements in the field, as it seems there will not be a simple relationship to be found between the variance slope for physical and biological variables, but rather only an underlying set of relationships between the physical environment and biological concentrations, and then from concentrations to variance slope. This is an important step toward improving biogeochemical and climate models and satellite data analysis in the future.

We thank all three reviewers for a thorough and very constructive review. Their review helped us significantly improve and clarify the manuscript. All comments have been addressed below.

Please also note that we found and corrected a small error in the calculation of variance slope. Specifically, we were calculating from ~3km to ~75km instead of ~0.5 to ~75km as stated in the text. We have corrected this to calculate from 0.5 to 75km which has changed some of the slopes slightly and this has been updated in all figures and text.

Reviewer #1 (Remarks to the Author):

Summary: The authors perform a global analysis to show that some biological and physical parameters are decorrelated across spatial scales. Although this analysis is fairly comprehensive and relies on extensive data, I have several concerns with the data interpretation and the robustness of the results. With some extensive modification, I could see this manuscript being a useful and timely contribution.

Major concerns:

1. The terms 'ocean biology' and 'ocean physics' are far too broad for the data that are actually being studied. There are many components of ocean biology, such as phytoplankton diversity, physiological traits, etc. that are not directly captured in this study. Similarly, the influence of ocean physics goes beyond temperature and salinity (wind speed, ocean currents etc.). I would suggest that the authors limit their title, abstract and major conclusions to prevent over-interpretation of the data actually being studied. In the best-case scenario, this study is arguing for a decorrelation (not decoupling) between phytoplankton biomass and seawater density, not ocean physics and biology.

We have limited the scope of the title and terminology. It is a good point that this isn't necessarily a decoupling but rather a lack of correlation. We have clarified further that we aren't arguing for a decorrelation of biomass and density, but rather their cascades of variance. We have changed the title to now say plankton instead of biology and physical properties instead of physics to be more precise and in line with previous studies which have used these same parameters.

2. Although I understand the reasons for using linear regression to identify the strength of the relationship, the distribution of points for many of the plots shows that the fit is not optimal. The effect sizes are also low. In Fig. 4, the greatest R² value is 0.46 (between Γ_{density} and Γ_{temp}), which makes sense, given the known relationship between seawater density and temperature. Otherwise, there are no additional analyses to support the arguments throughout the paper. In its current form, I am not entirely convinced that the data support the authors' conclusions (Ln 226-235).

We have clarified that we are testing if patchiness in one field is explained by the other, following on previous work examining these relationships. We have also noted that the

calculation of Γ is non-parametric and that much of what drives these relationships in the presence of noise is their dynamic range relative to subpixel variability. Note that despite Γ having a very small dynamic range (the range of Γ is from 0 to 1.5), it shows robust relationships consistent within the physical or biogeochemical parameters. This dynamic range is significantly smaller than for any of the variables themselves.

Here are some suggestions:

- It might be interesting to highlight where Γ_{chl} and Γ_{temp} are equal, where Γ_{chl} exceeds Γ_{temp} and where Γ_{chl} is less than Γ_{temp} on a map. This could add nuance to your arguments and provide more context for the results.

Following this comment, we have added a plot as suggested into the supplemental (Figure S4) and discuss it in the results.

- Consider reinforcing the correlation plots with supplementary figures showing what the residuals look like.

We have generated these plots, but they all generally cluster around zero, with no structure suggesting the linear relationship is appropriate.

- Density plots could indicate where the points are concentrated when comparing any 2 variables. The Longhurst provinces all seem to cluster around the middle, with some interesting exceptions.

Thank you for this suggestion, we have tried a number of options to better show the density of the points and have now added contour lines to show the density of points which does indeed make these plots more interpretable.

3. Ln 261-270: All of Fig.6 just seems to show clouds of points with no obvious correlation. The regression lines also seem weird (6a-c) compared to the reported equations.

We have confirmed that the regression lines are all correct as written. The density contours described above also help clarify the relationships.

- Why are the intercepts so different in the equations (~ 0.7)? The lines on the plots show an intercept less than 0.6.

We have re-confirmed in analysis and in the figures and these intercepts are all ~ 0.7 and in each plot at $x=0$ the intercept matches the reported value. Because the plots all go into the negative on the x-axis (because it is a log plot) we understand this may be confusing.

- In 6a, why does Γ_{chl} only go up to 0.9?

Because nearly all the Γ_{chl} values fall from 0 to 1 we limited it to better show the data. We have changed this upper limit to 1 instead of 0.9 to span nearly all the data.

- In 6c, why does the line on the actual plot appear to show positive slope when it is -0.21?

This parameter is inversely related to size so we had initially flipped the x-axis to better show the same pattern across variables, but we understand this may be not clear to readers so we have flipped it so they are all consistent.

These are some major irregularities.

4. Ln 279-283: O₂ saturation is not a nutrient. In this case as well, I am not sure why Γ_{chl} only goes up to 0.9 when Fig. 2 shows the histogram going up to 1.5.

Thank you for the catch on not clarifying O₂ saturation specifically in the caption. This has been corrected. Γ_{chl} was limited to 0.9 to focus on the Longhurst province data distribution and the bulk of the data. We have changed this plot to a max of 1.0 which spans nearly all the Γ_{chl} data.

5. Some additional thoughts and questions:

- From what I understand, the data presented here were all measured right at the surface. It might be useful to discuss whether vertical mixing rates, phytoplankton mobility, buoyancy (species composition) could alter observed patchiness on short time scales.

- I am surprised there was no mention of local wind patterns on observed chlorophyll patchiness, especially given that the sampling was conducted on a sailing vessel (Ln 413). Instead of temperature and salinity, my understanding was that wind, currents and mixed layer depth tend to be important in determining patchiness.

George, D. G., & Edwards, R. W. (1976). The Effect of Wind on the Distribution of Chlorophyll A and Crustacean Plankton in a Shallow Eutrophic Reservoir. *Journal of Applied Ecology*, 13(3), 667–690. <https://doi.org/10.2307/2402246>

Kahru, M., S. T. Gille, R. Murtugudde, P. G. Strutton, M. Manzano- Sarabia, H. Wang, and B. G. Mitchell (2010), Global correlations between winds and ocean chlorophyll, *J. Geophys. Res.*, 115, C12040, doi:10.1029/2010JC006500.

Thank you for these helpful references. We have clarified that we are revisiting previous work with the same variables (which did not include wind and other physical variables). Additionally, we are not looking at correlations between biological and physical parameters here, or even at correlations in their variance, but in their spatial patchiness. Atmospheric parameters like wind have different mesoscale bands (5 days, 100km) vs those of the ocean (1mo, 30km) and thus, a-priori, are not expected to have similar patchiness as the much more viscous and denser ocean. MLD variance is expected to be reflected in temperature (e.g. deepening mixed layers have anomalous low temperature and higher densities).

Overall, by generalizing over the entire dataset and not really looking into the actual spatial patterns, the conclusion that there is “a decoupling between ocean physics and biology” seems overstated.

While there are likely many interesting investigations into the specific details and patterns in each transect, here we were investigating statistical patterns in spatial organization over a large dataset. Towards clarifying this goal we have substantially toned down the language stating now in the abstract that “Here we show that the patchiness of physical parameters and plankton are uncorrelated across the globe” and further in the results that our observations “suggest that biological processes may modulate the initial physically-driven spatial pattern baseline sufficiently to manifest as uncorrelated levels of patchiness”.

Additional comments:

Fig. 2 (right panel): Might be useful to indicate the Γ values more explicitly for readers. Showing the entire equation is fine, but I expected some discussion on the intercept values (the fourth panel looks different). It is also interesting to me where the red and green lines cross (at what length scale). Also, consider using a color-blind friendly combination.

We have added a note that the variance slope can be seen in the legend for each panel and that the intercept value corresponds both to the magnitude of the variance and the slope. For colorblind friendliness we have used a dashed line and one of our authors who is colorblind has noted it is clear for him.

Fig. 3: Why is the range of the distribution greater than the legend on the right? It seems the values for Γ_{chl} go up to 1.5 and Γ_{temp} go up to 1.9, but the legend ends at 1 and 1.5.

This was to emphasize the bulk of the distribution of the data, which for Γ_{temp} falls within 0.5 to 1.5 and for Γ_{chl} falls within 0 to 1. This allowed us to show them both with the same span and more clearly show the bulk of the data.

Ln 239: “Marginally more correlated” is quite the bold claim. When comparing an R^2 of 0.04 vs 0.03, I think it is safe to say that Γ_{chl} and Γ_{temp} are not correlated in both cases.

This is a good point, we have removed this statement and stated that “the variables are still effectively uncorrelated”.

Fig. S6: Please check the equations for the last 2 panels.

Checked. These equations are correct as written.

Reviewer #2 (Remarks to the Author):

Review Gray et al.

This paper examines a large data set of underway, near-surface, bio-optical, temperature and salinity data from the S/V Tara. The data set and the information it holds is impressive, as the ship's track covers the Atlantic and Pacific, with multiple basin crossings, as well as tracks along the basin margins. The authors examine the patchiness of physical (temperature and salinity) and biogeochemical (chlorophyll-a, beam attenuation $cp(443)$ and γ .) variables using the exponent of the variance vs length-scale (V-L) relationship for each of the variables. The exponent can be used as a patchiness measure, as shown previously in ref. 31.

Consistent with previous literature, the authors find that the patchiness (exponent of the V-L relationship) differs between physical and biogeochemical variables. They also find that the patchiness of physical variables is not correlated with the patchiness of biological variables across different regions of the oceans. This leads them to conclude that "patchiness of physics and biology are decoupled at the global scale." This conclusion is misleading. On first reading, it would be easily mis-understood that the authors conclude that patchiness of biological variables is not related to the patchiness of physical variables. This would be wrong. The physics does indeed affect the patchiness of biology. I think that the authors' conclusions are either incorrect or mis-stated.

This is a helpful point and clarification. We do not mean to say that they are unrelated, but rather biological patchiness is not statistically correlated to physical patchiness in our dataset. So not necessarily decoupled, but that the variance biological patchiness cannot be explained by the physical patchiness. Our observations suggest plankton are active tracers in the sense they have an impact on their spatial organization, physics creates the baseline and our results suggest biology modulates the patchiness sufficiently to decorrelate. We have changed the title, language, and added more details to clarify this point. We have also added a Figure (S4) which describes locations where the chlorophyll-a is patchier than temperature, less patchy, and equal to show further nuances of this across the globe. Based on this suggestion we have also conducted a new analysis of patchiness for CDOM, which we collected for the second half of the cruises used in this paper, and this indeed seems to fall somewhere in between the patchiness of physical and biological variables. CDOM can be thought of as a somewhat more passive tracer, following physical mixing more closely but also slowly responding to biological processes. This is shown in Figure S8.

Assuming that the conclusions (interpretation) is mis-stated — it would be worth understanding why the authors expect the exponent of physical and biological variables to be correlated. Even if the distribution of biological variables were shaped by the physics, the difference in the exponent of the V-L relationship could vary depending on the time scales of the biological processes as compared to the physical processes. Their ratio could differ regionally and hence the findings should not lead to the conclusion that the authors have drawn.

Our work here is the result of a data-driven exploration, given that the fields are strongly correlated, and so are their variances, we might expect their patchiness to be correlated, but our findings are that their patchiness values are uncorrelated. Beyond this baseline if we think of plankton as a quasi-passive tracer in an environment where advection dominates, we might

expect the patchiness of physical parameters and plankton to be similar. We are investigating instantaneous patchiness, but as pointed out there could be a lagged correlation in patchiness and we have noted that this could account for some of the lack of statistical explanation.

There are several other issues (inaccuracies) in this paper. The term “global scale” is used incorrectly. The authors mean to convey that the relationship between the regional-scale patchiness of temperature and chlorophyll-a differs from one region to another. This result is not surprising nor does it challenge any of the previous literature that states that the patchiness of biogeochemical and physical variables is related. The result does not pertain to the global scale (largest scales).

Thank you for this helpful point. We no longer write ‘global scale’ in the paper, and instead we’ve clarified that across all the studied environments, given our analysis window is 0.6 - 100km, we find that patchiness of physical parameters and plankton is uncorrelated.

Please see: Campbell, J.W., Lognormal distribution as a model for bio-optical variability in the sea, JGR 1995, which addresses some of the questions raised in the manuscript.

Thank you for this helpful reference that indeed has an interesting discussion on many points raised in the manuscript. We have added a reference to this and included some points in the discussion.

What exactly is “gamma”? The paper refers to this only once and explains it as an optical proxy for mean particle size.

Gamma (γ) is an optical size parameter that is inversely related to the mean size of particles from approx 0.2 μ m to 20 μ m. It is called gamma in optical oceanography but we have removed this from the manuscript to avoid confusion with Γ and instead refer to it as mean particle size parameter (γ). It is derived from the beam attenuation coefficient of particles and has been shown to correlate well with the slope of the particle size distribution, e.g. Boss et al., 2001, Applied Optics.

Methods: In the text, the authors say the spacing between measurements is about 150m. But since the measurements are at certain time intervals, the lateral spacing between 2 measurements could differ considerably. The authors do the analysis using windows defined by time (lines 414-415). This could affect the results significantly. It is just like calculating spectra for unevenly spaced data, assuming equal spacing. At the least, the authors need to show the spacing - and the variability in the spacing of measurements.

We have shown that random perturbations in spacing don’t materially influence the slope which is quite robust to these small differences. We also did an extended analysis to determine the ideal minimum spacing between two measurements and we found the results were not sensitive to the minimum lateral distance range used in this analysis (100m to 300m). We have added a histogram of the median spacing between each sample per transect as a supplemental figure

(Figure S10). These attributes are a primary benefit and motivator for using this specific method over the classical Fourier based power spectra calculation and why we have used it here for this dataset.

Supplementary materials was not very helpful because the figure captions were inadequate and there was no supporting text.

We have extended the discussion in the captions of the supplemental materials and added further supporting text.

Line 23: slope of Variance vs. Spatial scale (rather than spatial scale vs. variance)

Corrected.

Line 27: Incorrect to state that the patchiness of physical and biological parameters are uncorrelated.

We have clarified that physical properties and plankton patchiness are uncorrelated based on our analysis, though they may be correlated across subsets of the data. We have also further explained that their absolute values and their variance may be strongly correlated. We have emphasized that this does not mean that physical processes don't impact biological spatial patterns.

Line 28 What is meant by "variance slope is an emergent property"?

By emergent property we mean a property that comes from behavior of the ecosystem (such as prey-predator interactions and reproduction) and the impacts of physics - a property that emerges from the ensemble of processes in the ecosystem and environment. We have now defined this in the manuscript.

Line 30 No, this does not "provide context for decades of discrepancy"

We have changed this to now state our results provide context for decades of observations with different interpretations.

Line 31 New tests? What is meant by this?

Here we mean using spatial patchiness as an assessment for models, to see if their spatial patchiness distributions and relationships are similar to those found at the same scales in observations. We have clarified this further in the discussion.

Line 32 insight instead of "new insight"

Corrected.

Line 40 that result in

Corrected.

Line 43 “with close proximity” What does this mean?

Here we mean that the ocean can contain different patches near each other that can have very different physical and biogeochemical properties and that this supports biodiversity.

Line 49 Is there a reference to substantiate this statement? “generally increasing patchiness as one moves ups the trophic chain”

This is addressed in both of our cited examples, for example section 4.4 in Estapa et al. We have added an additional citation to one of the original papers discussing this:

Mackas, D. L., and C. M. Boyd (1979), Spectral analysis of zooplankton spatial heterogeneity, Science

Line 72 “this parameter” — Spectral slope?

Correct, this has been clarified.

Line 73 replace “thought of as quantifying” by “interpreted”

Changed.

Line 75 It is not a decomposition

We have changed decomposition to analysis.

Lines 78-90 —There are many incorrect or inaccurate statements in this section.

We have corrected each noted point below, thank you for these suggestions.

Line 79 Energy cascades, not turbulence

Corrected

Line 79 Replace “turbulent mesoscale” by “mesoscale” as the large scale turbulence is different from 3D turbulence

Corrected

Line 81 No, this is not how it works. Energy is transferred up scale at small Rossby number (the downscale cascade is facilitated by submesoscale processes)

Here we have summarized and updated the description in the text based on the overview of turbulence in Atmospheric and Oceanic Fluid Dynamics, Vallis, 2017 2nd edition.

Lines 84-87 This is confusing and misleading. The inertial subrange of Kolmogorov is different from the large-scale dynamics of the ocean - which is affected by rotation and stratification.

Here clarify the 2D - large scale from 3D inertial subrange turbulence.

Line 88-90 Temperature could have a variance source at smaller scales too — for example, due to upwelling on small scales.

This is a good point and we have added this in this paragraph.

Line 92 Previous literature did not attribute phytoplankton patchiness simply to turbulent stirring

We have changed this to say that phytoplankton patchiness was initially thought to be primarily controlled by turbulent stirring.

Line 103 These references (7,30,31) did not all corroborate each other — on the contrary, they gave different reasons for the observed patterns in patchiness of temperature and chlorophyll.

Corroborate was an imprecise word, thank you for catching this. We have changed this to “come to a similar conclusion”. While these three studies have different specific conclusions Abrahams states:

- “The characteristic spatial patterns of the phytoplankton and zooplankton are a consequence of the timescales of their response to changes in their environment caused by turbulent advection.”

And Mahadevan notes:

- “This suggests that sea surface Chl is more patchy (has smaller p) than SST at mesoscales because the characteristic time scale of phytoplankton growth in response to the availability of nutrients is less than that for the equilibration of temperature in response to heat fluxes.”

Since Levy and Klein is not directly commenting on this same aspect, we have removed this citation in this line and discuss it later in this paragraph in relation to their discussion of physics playing a dominant role in the spatial variability of plankton.

Line 105 What is the “new” heat flux?

This was imprecise language and “new” has been removed.

Line 108 The SST equilibrates with the atmosphere largely due to air-sea heat exchanger and the statement “heat is mixed over time” is not correct.

This has been corrected.

Line 109 “return to covarying” - not correct

This has been corrected to say that the patchiness in these variables returns to co-varying.

Lines 103-113 The references are inaccurate

As noted above we have changed these citations, moving the reference to Levy and Klein and confirmed the others apply directly where noted.

Line 115 What is meant by “actual observations remain inconsistent” ?

This has been re-written to say that the interpretations from observational studies remain inconsistent.

Lines 116-119 - could use a better word than “work” which appears repeatedly

Changed to “studies” and “investigation”.

Line 120 “yet not so above 5 km to 80 km” what does this mean?

This has been re-worded to clarify the scales being discussed.

Line 142 Measurements of what?

These are outlined in the following sentence, we have clarified the text here to connect these sentences.

Line 173 (fig 2 caption) What are “absolute” values

This was to differentiate between measured values and variance but we see it could be confusing and have removed “absolute”.

Line 194 — related to the reference by Campbell (see above)

We have shown that the metric used is not sensitive to the log distribution of chl_a or the attenuation coefficient, and that the calculated $\int \text{chl}_a$ is identical whether chl_a is log transformed before being input into the calculation.

Line 206-208 - previous papers have not reported contradictory results — not sure why the authors find contradictory results when using satellite data

Yes precisely, this contradictory result is a key finding of this paper and a major issue if satellite data is being used for a similar method. We have not found other papers that reported this contradiction.

Line 214-216 Could this be because chlorophyll is log normally distributed

We have tested the algorithm before and after a log transformation and it is insensitive, i.e. non-parametric, to the distribution of the data, and have added a sentence to that effect in the text

Line 234-235 - No. This conclusion is not substantiated by the findings.

We have re-worked this conclusion and defined emergent. This section now states that plankton patchiness is modulated by biological processes sufficiently that the variance in patchiness cannot be statistically explained by the concurrent physical patchiness field. By emergent property we refer to a possibly unexpected outcome from the fields, a property that comes from behavior of the ecosystem (e.g. prey-predator interactions, reproduction, viruses). We have clarified this in the manuscript at the first mention of 'emergent'.

Fig 7. Same as Fig 6a except for the color of dots?

Correct, this is showing the same data but colored by nutrient concentration in the provinces.

Fig 8. Could this figure go in the supplementary material?

We consider Figure 8 critical to the conceptual understanding of the patchiness we observe and for showing that this fine scale variability stems from high quality measurements not from noise in the instrument or measurement process.

Line 315 scale

Corrected

Line 320 - seems that Reference 31 is for oligotrophic regions, whereas the results presented here are for productive regions

Our study area includes many oligotrophic regions (e.g. subtropical gyres in the Pacific) in addition to more productive waters.

Line 353 - "physical patchiness is more random" - this does not seem correct

We have changed this to state that physical patchiness is less coherent spatially and organized differently. We have quantified this via the autocorrelation of patchiness at multiple lags, where the autocorrelation of $\square_{\text{chl}a}$ is higher than \square_{temp} at all lags. We have also added this to the results.

Line 361-2 why should there be an interdependence?

If plankton were simply a passive tracer we could expect a strong relationship between the patchiness of plankton and temperature.

Line 365 - this is not a conclusion that can be drawn from the results presented - and seems incorrect

We have removed the word "independently" and reworked this whole section.

Due to the incorrect interpretation of findings, several inaccuracies, and understanding gaps of previous literature, I do not recommend publication of this article.

We have addressed each point provided, all of which we believe have considerably clarified and strengthened the paper. We thank you for your thorough review and time spent on this work.

Reviewer #3 (Remarks to the Author):

This article examines the scales of variability in both optical and physical variables from in-situ ocean surface observations. Results demonstrate that basin-scale variance slope does not correlate between the biological/biogeochemical and physical variables. A shorter analysis of similar variables from satellite data (mainly in the supplement) demonstrates one reason why previous studies have found correlations in spectral slopes or variance slopes, as those do occur in the satellite record. These results are critical for understanding decades of disagreements in the field, as it seems there will not be a simple relationship to be found between the variance slope for physical and biological variables, but rather only an underlying set of relationships between the physical environment and biological concentrations, and then from concentrations to variance slope. This is an important step toward improving biogeochemical and climate models and satellite data analysis in the future.

This is a well-written article with clear methods and results. The analysis supports the conclusions drawn. There are no major flaws. I recommend this article be returned for minor revisions to address comments below.

Minor Revisions

1. Line 120, citation 17. Is the correlation below 5km and maybe above 80km, or is the correlation occurring at small scales with an uncertain upper limit between 5 and 80 km? Please rewrite for clarity.

We have re-written this to be clearer in both instances where this reference occurs in the manuscript.

2. At the beginning of the results, please repeat or move (from line 198) the equivalent value for Gamma (variance slope) to $-5/3$ for spectral slope. I really wanted to know at that time when reading.

We have repeated the variance slope value that corresponds to $-5/3$ in this first results paragraph for easier reader interpretation.

3. In Figures 1 and 3, the dots on the map are quite small. Is it possible to increase their size while still allowing the gaps to be visible? Maybe they can be drawn as ellipses or rectangles so that there are larger colored areas without excessive overlap.

We agree the dots are small. But given the legs in close proximity it is challenging to make them larger without covering other data. These maps are very high resolution and we will ensure if published that the highest resolution possible is shared so that readers can zoom in and examine the data clearly.

4. Line 230: please add a brief description of Longhurst provinces and note that Figure 5 shows their location-- I read this as the averages would be in Figure 5 separately from the full data in Figure 4.

Brief description added as well as a note that Fig 5 shows their location.

5. Figure 6c: Please add a note as to why the x-axis is reversed or remake this with an increasing x-axis. It is quite jarring to see a negative slope in the labeled equation with the visually 'positive' slope. Also please explain why $R^2=0.02$ is considered correlated and $R^2=0.00$ is not. Even though p is small, it is hard to understand why this is convincing. Would showing the log be better?

We have changed this back to not be flipped on the x-axis for consistency. This is a good point about the R^2 value. In the text we specify that only the concentration dependent variables (chl-a and cp) have a correlation between patchiness and absolute value.

6. Figure 7d: The Longhurst province averages for O₂ do not appear to be correlated to chl-a or Gamma_chl-a here. Please confirm that this is the case, as you assert (line 282) that this is a clear increasing relationship and by eye it is not.

We have corrected this to say "Oxygen saturation has a slight decreasing relationship with more considerable outliers."

Typos Etc.

1. Line 290: add 'in' after 'zooming'

Corrected

2. Figure 8: please label the location this data is from in more detail-- can you give a latitude/longitude for the middle, perhaps?

We have now noted the centroid for this transect which is centered on 20.23 S, 152.90 E.

3. The end of the results section includes author names for references, which is not the case elsewhere. Please double check these choices.

Corrected

4. In the references, sometimes 'Boss' does not have initials for first/middle name.

Thank you for this detailed check, we have corrected author initials.

5. Please check references to make a consistent format. The number of authors included, for example, before going to et al., is variable.

We have corrected references to a consistent format.

This email has been sent through the Springer Nature Tracking System NY-610A-NPG&MTS

REVIEWERS' COMMENTS

Reviewer #1 (Remarks to the Author):

I think the revised text and figures more closely align with the stated claims in the manuscript. My only suggestion would be to perform a careful proofreading, as I noticed some minor grammatical errors and miscitations.

Thank you for your follow on review, we have gone through the manuscript more thoroughly and corrected typos and errors in grammar as well as cleaned up the figure captions for clarity, proper format, and language consistency.

Reviewer #3 (Remarks to the Author):

This is a well-written article with clear methods and results. The analysis supports the conclusions drawn. There are no major flaws. The revised manuscript has addressed all my previous minor concerns. I recommend this article be published in this journal.

Thank you for this review, we appreciate your prompt second round of comments.

Broader description:

This article examines the scales of variability in both optical and physical variables from in-situ ocean surface observations. Results demonstrate that basin-scale variance slope does not correlate between the biological/biogeochemical and physical variables. A shorter analysis of similar variables from satellite data (mainly in the supplement) demonstrates one reason why previous studies have found correlations in spectral slopes or variance slopes, as those do occur in the satellite record. These results are critical for understanding decades of disagreements in the field, as it seems there will not be a simple relationship to be found between the variance slope for physical and biological variables, but rather only an underlying set of relationships between the physical environment and biological concentrations, and then from concentrations to variance slope. This is an important step toward improving biogeochemical and climate models and satellite data analysis in the future.